

# Soil and forest floor carbon balance in drained and undrained hemiboreal peatland forests

Aldis Butlers[1], Raija Laiho[2], Kaido Soosaar[3], Jyrki Jauhiainen[2], Thomas Schindler[3], Arta Bārdule[1], Muhammad Kamil-Sardar[3], Andreas Haberl[4], Valters Samariks[1], Hanna Vahter[3], Andis Lazdiņš[1], Dovilė Čiuldienė[5], Kęstutis Armolaitis[5], Ieva Līcīte[1]

[1]Latvian State Forest Research Institute (Silava), Salaspils, 2169, Latvia
[2]Natural Resources Institute Finland (Luke), Helsinki, P.O. Box 2, 00791, Finland
[3]Department of Geography, University of Tartu, Tartu, 51014, Estonia
[4]Michael Succow Foundation partner in the Greifswald Mire Centre, 17489 Griefswald, Germany
[5]Department of Silviculture and Ecology, Lithuanian Research Centre for Agriculture and Forestry, Kėdainiai distr., 58344, Lithuania

*Correspondence to*: Aldis Butlers (aldis.butlers@silava.lv)

**Abstract.** Drainage of organic soil is associated with increasing soil carbon (C) efflux, which is typically linked to losses in C stock. In previous studies, soil in drained peatland forests has been reported as both a C sink and source depending on, e.g., soil nutrient and moisture regimes. However, most of the earlier research was done in boreal sites, and the impact of soil moisture regime on soil C stock is likely to vary across different climatic conditions and ecosystems, depending further on vegetation. In this study, we examined the soil and forest floor (including ground vegetation) C balance in drained and undrained hemiboreal forests to evaluate drainage impact on C balance. A two-year study was conducted in 26 drained and undrained forest stands with nutrient-rich organic soil in the Baltic states (Estonia, Latvia, Lithuania). To assess the C balance, measurements of soil heterotrophic and total respiration were carried out, along with the evaluation of C influx into the soil through litter, including fine foliar litterfall, herbaceous ground vegetation, and fine roots of trees. The $CO_2$ emissions did not significantly differ between the study countries; therefore, one emission factor can be applied to characterize soil emissions in the Baltic States. It was observed that C influx into the soil through litter can compensate for the C losses caused by heterotrophic soil respiration, and neither drained nor undrained soils were proven to be losing their C stock. Comparing the C balances in drained and undrained sites, it was found that drainage of organic soils reduces their C sequestration by $0.43\pm2.69$ t C ha$^{-1}$ year$^{-1}$.

## 1 Indtroduction

Soil in peatlands, characterized by its high content of partially decomposed plant matter, is a major terrestrial organic carbon (C) stock, estimated to range from 504 to 3000 Gt C (Scharlemann et al., 2014). Although northern peatlands make up only 2-4% of the global land area, they contain a substantial amount of soil C, ranging from 126 to 621 Gt C (Yu, 2012), highlighting the significance of these peatlands in the global C budget. About 28% of the pristine (undrained) peatlands globally are inherently covered by forest (Zoltai and Martikainen, 1996), and those forested peatlands in the boreal biome can accumulate C into the soil at similar rates to non-forested peatlands; the higher decomposition rates observed in forested peatlands (Beaulne et al., 2021) can be compensated by higher litter inputs (Straková et al., 2011). To enhance wood biomass increment, peatland drainage for forestry purposes has been commonly applied in the past. Drainage facilitates oxygen access to deeper peat layers, thereby promoting tree root survival and function, but also the mineralization of organic matter and the release of C into the atmosphere in the form of $CO_2$. Therefore, the conservation of organic soil C stocks in managed peatlands has attracted attention in the context of climate change.



The approximately 13 million ha of forestry-drained organic soils in Europe have been estimated to emit 17 million tons of
$CO_2$ per year (Pilzecker et al., 2022). In the Baltic states (Estonia, Latvia, and Lithuania), the total area of drained organic
forest soils is reported to be 0.8 million ha, with emissions of 1.8 million tons of $CO_2$ per year (Ministry of the Environment
of Republic of Estonia, 2021; Konstantinavičiūtė et al., 2023; Skrebele et al., 2023). Thus, countries with a relatively small
total land area yet a considerable proportion of organic soil can have a considerable role in organic soil management. This
underscores the importance of acquiring precise estimates for the impact of organic soil drainage on $CO_2$ emissions in this
region.
The Baltic States are located next to each other in the hemiboreal vegetation zone (Ahti et al., 1968) – halfway between the
temperate and boreal zones – and thus, similarities in soil $CO_2$ emissions may be expected. However, the emission estimation
approach is currently not harmonized as the countries use different emission factors to estimate emissions (Ministry of the
Environment of Republic of Estonia, 2021; Konstantinavičiūtė et al., 2023; Skrebele et al., 2023). A similar issue can also be
observed in a broader geographic scale, leading to problems of comparability of estimated emissions within and between
different climate regions, as emission factors are best suited for application in geographic areas that share similar conditions,
rather than being bound by country borders.
Guidelines of Intergovernmental Panel on Climate Change (IPCC) intends to address anthropogenic greenhouse gas emissions
(Eggleston et al., 2006). Therefore, when evaluating human-induced emissions, it should be a good practice to consider the
natural background emissions as well. In the context of organic soil drainage, the corresponding emissions should be expressed
as difference between emissions from undrained and drained soil, rather by expressing direct emissions from drained soils. For
this reason, in inventories the off-site $CO_2$ emissions are evaluated by comparing leaching of dissolved of organic C in
undrained and drained organic soils. However, while the IPCC guidelines aim to address the impact of drainage on $CO_2$
emissions from organic soil, data limitations hinder the elaboration of such default emission factors (EF) for on-site $CO_2$
emissions. As a result, for elaboration of default IPCC EF study results on $CO_2$ emissions from drained soils are compiled.
According to National Greenhouse Gas Inventories submissions of 2023, the $CO_2$ emissions of drained organic forest soil in
the Baltic states, except Latvia, were estimated using the default EF provided by IPCC for the temperate region (Calvo Buendia
et al., 2019). Currently only one default IPCC EF for the whole temperate climate region is available and it does not involve
any data measured in the Baltic states (Hiraishi et al., 2013a). EF is elaborated using results from 8 sites with drained soil
(Hiraishi et al., 2013a) published in 5 articles (Glenn et al., 1993; Minkkinen et al., 2007; Yamulki et al., 2013; Von Arnold et
al., 2005b, a) on studies representing a wide climatic gradient and different $CO_2$ estimation methods, which further complicates
the comparability of the results that have been aggregated (Jauhiainen et al., 2019, 2023). A recent synthesis study evaluated
whether default IPCC EF can be improved by compiling results from most recent studies. Still, only modest, and insignificant
changes judging by confidence intervals of IPCC EFs could have been introduced for the temperate climate region [16]. This
was because both the number and the geographical representation of studies of drained soil done in the temperate zone is still
scarce and does not enable further stratification of site conditions within the region. Recognizing that additional data on
undrained soils are necessary for assessing the net impact of drainage on $CO_2$ emissions, the knowledge on drainage related
organic soil $CO_2$ emissions is poor. In the few studies on drained and undrained soil C balance conducted in the Baltic states,
using both chamber and soil inventory methods, findings have been inconsistent (Vigricas et al., 2024; Butlers et al., 2022;
Lazdiņš et al., 2024; Bārdule et al., 2022). Organic soils have been identified as both C sinks and sources, with no decisive
conclusions reached regarding the factors driving such variation. This indicates the need for continued efforts to conduct local
studies to fill the knowledge gaps on organic soil $CO_2$ emissions in Cool Temperate Moist climate region (Calvo Buendia et
al., 2019) overlapping with hemiboreal vegetation zone.
In this study, we evaluated the nutrient-rich soil and forest floor C balance in drained and undrained hemiboreal peatland
forests with different tree species in Estonia, Latvia, and Lithuania. The aim was to quantify the impact of drainage on $CO_2$
emissions by comparing soil and forest floor (including ground vegetation) C influx and efflux in drained and undrained sites.




For this purpose, research was carried out in 26 forest stands over two years, analyzing forest floor $CO_2$ emissions and C inputs
by tree fine roots, ground vegetation, and fine foliar litter. We hypothesized that in the Baltic states, consistent emission factors
can be used to estimate organic soil $CO_2$ emissions, whether they are tailored to the dominant tree species or applied as a single
factor across all forest lands. Results acquired can provide empirical data for future syntheses aiming to elaborate static or
dynamic emission factors.

## 1 Materials and methods

### 1.1 Study sites

In total, 26 study sites (Figure 1) were established in stands dominated by black alder (*Alnus glutinosa* (L.) Gärtner), birch
(*Betula pendula* Roth, *Betula pubescens* Ehrh.), Scots pine (*Pinus sylvestris* L.), and Norway spruce (*Picea abies* (L.) Karst.)
of different ages (Table 1). The study sites included both drained (n=19) and undrained (n=7) soils, with the peat layer thickness
ranging from 27 cm to over 2 meters. Soil drainage status was determined based on the presence of drainage ditches along the
forest stand borders. According to forest type classification, all of the sites were characterized as nutrient-rich (Bušs, 1981).

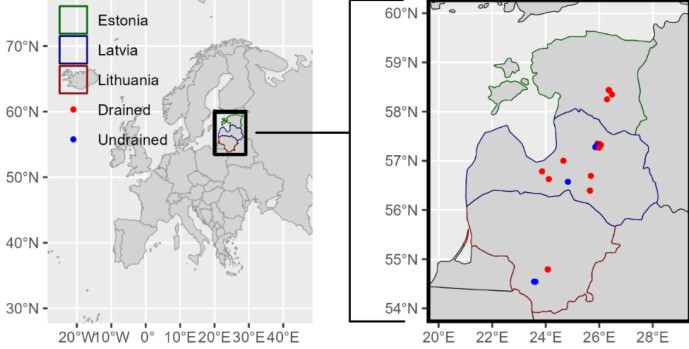

**Figure 1: Locations of the study sites.** Points indicate the locations of study site clusters.
Despite the more numerous drained sites showing greater variation than the undrained sites, overall, the two groups of sites
had similar stand characteristics (Table 1). The mean stand age of both groups was 74 years, with a range of 26-162 years for
drained and 44-96 years for undrained sites. Average basal areas in turn were 27 for drained and 30 m² ha⁻¹ for undrained sites,
respectively. Detailed information on stand characteristics, including mean soil water table level (WTL) and coordinates, is
provided in Table S 1.

**Table 1: Summary of tree stand characteristics in the study sites.**

| Parameter | Dominant tree species | | | | | | | |
|---|---|---|---|---|---|---|---|---|
| | Black alder | | Birch | | Pine | | Spruce | |
| | Drained | Undrained | Drained | Undrained | Drained | Undrained | Drained | Undrained |
| Site count | n = 2 | n = 2 | n = 5 | n = 2 | n = 5 | - | n = 7 | n = 3 |
| Age, year | 30…80 | 44…74 | 24…45 | 44…61 | 60…141 | - | 40…162 | 81…96 |
| Mean height, m | 13…20 | 16…28 | 13…18 | 9…20 | 12…21 | - | 10…23 | 15…20 |
| Mean diameter, cm | 12…21 | 16…28 | 12…22 | 8…21 | 12…22 | - | 10…25 | 17…21 |
| Basal area, m² ha⁻¹ | 26…36 | 30…36 | 15…23 | 22…23 | 17…48 | - | 18…36 | 25…42 |




The annual mean air temperature during the study period varied from 6.4 °C in Estonia to 8.5 °C in Lithuania, while the annual
precipitation in the whole region was 600.9±76.7 mm (Table S2) (Estonian Environment Agency. Climate normals, 2024;
Latvian Environment, Geology and Meteorology Centre. Climate normals, 2024; Lithuanian Hydrometeorological Service.
Climate normals, 2024). The long-term (1990-2020) annual mean air temperatures in Estonia, Latvia and Lithuania were
6.4 °C, 6.8 °C and +7.4 °C, respectively, while the mean annual precipitations were 662 mm, 686 mm and 695 mm,
respectively (Estonian Environment Agency. Climate normals, 2024; Latvian Environment, Geology and Meteorology Centre.
Climate normals, 2024; Lithuanian Hydrometeorological Service. Climate normals, 2024).
At each study site, three sampling locations (subplots) were selected, ensuring a minimum separation distance of 15 m along
a transect. Transects were positioned perpendicular to drainage ditches in drained areas and perpendicular to stand borders in
undrained areas. In the drained sites, the first subplot was located five to ten m from the nearest drainage ditch.
Empirical data was gathered from January 2021 to December 2022 in Estonia and Latvia, and from July 2021 to June 2023 in
Lithuania. The sites were visited monthly in Latvia and Lithuania, and biweekly in Estonia.
**1.2 Total respiration**
Forest floor respiration (including ground vegetation) further defined as total respiration (Rtot) measurements included both
soil heterotrophic respiration and autotrophic respiration of aboveground and belowground parts of ground vegetation. In the
article Rtot is equivalent to forest floor respiration, including respiration of ground vegetation. Gas samples were collected
from manual closed static opaque chambers (PVC, volume 0.0655 m$^3$) as described in the literature (Hutchinson and
Livingston, 1993) for subsequent laboratory analysis. Five to six ring-shaped chamber collars (area 0.196 m$^2$) were
permanently installed in the soil at a depth of five cm in each study site at least one month prior to the first sampling to avoid
the installation effect on fluxes. The soil surface and vegetation were kept intact throughout the whole flux monitoring period.
Thus, the Rtot measurements include $CO_2$ emissions caused by the decomposition of litter and autotrophic respiration of
ground vegetation plants included in the collar and the chamber headspace during the gas sampling.
The gas samples were collected during a measurement campaign by obtaining four samples from each chamber in pre-
evacuated (0.3 mbar) glass vials (100 cm$^3$). During the sample collection, the air within the chamber was not mixed, and
samples were taken from the sampling tube inserted approximately at the center of the chamber. The first sample was taken
immediately after placing the chamber on the collar, following the removal of dead volume from the sampling tube using a
syringe. Subsequent samples were taken at either 10 or 20 (Estonia) minute intervals over 30- or 60-minute monitoring periods,
respectively (Vigricas et al., 2024; Butlers et al., 2022).
The gas samples were analyzed using a Shimadzu GC-2015 gas chromatograph (Shimadzu USA manufacturing, Inc., Canby,
OR, USA) equipped with an electron capture detector (ECD). The uncertainty of the method used was estimated to be 20 ppm
of $CO_2$. Linear regression was applied to relate the $CO_2$ concentrations with the time elapsed since chamber closure for each
measurement. Subsequently, the measurement data was screened to identify deviations from the recognized trend, considering
the removal of measurements with identified errors. All measurements were discarded if the regression coefficient of
determination ($R^2$) was less than 0.9 (p<0.01), except for cases where the difference between the highest and lowest measured
$CO_2$ concentration in the chamber was less than the uncertainty of the method (specifically applicable during non-vegetation
periods).
The data that met the quality criteria were used to determine the slope coefficient of the linear regression, which was then used
to calculate the instantaneous Rtot according to the ideal gas law equation (Fuss and Hueppi, 2024):

$$Rtot = \frac{M \times P \times V \times slope}{R \times T \times A \times 1000} \qquad (1)$$

where Rtot is the instantaneous total respiration, mg $CO_2$-C m$^2$ h$^{-1}$; M is the molar mass of $CO_2$-C, 12.01 g mol$^{-1}$; R is the
universal gas constant, 8.314 m$^3$ Pa K$^{-1}$ mol$^{-1}$; P is the assumption of air pressure inside the chamber, 101.300 Pa; T is the air



temperature in the chamber, K; V is the chamber volume, 0.0655 $m^3$; slope is the $CO_2$ concentration change over time, ppm
$h^{-1}$; and A is the collar area, 0.19625 $m^2$.

**1.3 Soil heterotrophic respiration**

Heterotrophic soil respiration (Rhet) was measured by applying the manual closed dynamic nontransparent chamber method
(Denmead, 2008; Hutchinson and Livingston, 1993). For each measurement location, a 60 x 90 cm (W x L) trenched (Ngao et
al., 2007) plot was prepared at the end of the previous year's growing season to a depth of at least 40 cm, using geotextile on
the sides to prevent root ingrowth and by removing alive vegetation and litter layer. In each subplot, measurements were done
in 3 replicates, in total, nine measurement points in each study site. Every measurement lasted three minutes using the EGM-
5 portable $CO_2$ gas analyzer (PP Systems, Amesbury, MA, USA) and a fan-equipped chamber (area 0.07 $m^2$, volume 0.017
$m^3$) placed in the center of the trenched surface without using a collar. The measurement data was stored at a 1 Hz frequency.
Between the measurement campaigns, Rhet measurement points were covered with geotextile, which was covered with an
equivalent quantity of debris and litter as nearby soil, aiming to simulate natural conditions.
Before flux calculations, the first 15 seconds of the measurement data were discarded due to the potential error in the results
due to the placing of the chamber in the soil. To estimate the slope of the linear regression equation representing $CO_2$
concentration change in time, the same approach as for Rtot was used (Figure 1).

**1.4 Environmental variables**

Manual WTL measurements were carried out using nylon-mesh-coated, perforated piezometers (5 cm in diameter) in all
subplots. The lower end of the piezometer tube was installed at a depth of 140 cm. Also, manual soil temperature measurements
were carried out in all subplots, while continuous measurements - in the centermost subplot only. Manually, soil temperature
was measured at depths of 5, 10, 20, and 40 cm using a Comet data logger (COMET SYSTEM, s.r.o. , 756 61 Roznov pod
Radhostem, Czech Republic) equipped with Pt1000 temperature probes, and continuous measurements were carried out at
depths of 10 and 40 cm. Together with the manual soil temperature measurements, soil moisture was assessed at a depth of 5
cm using a ProCheck meter (Decagon Devices, Pulman, WA / USA) equipped with a moisture sensor GS3. All manual
measurements were carried out at the same time as $CO_2$ flux measurements. The continuous soil temperature measurements
with data loggers (Maxim Integrated DS1922L2F, iButtonLink Technology, Whitewater, WI 53190 USA) recorded values
every 30 minutes.
Soil samples were taken up to a depth of 75 cm at two locations in each subplot during the establishment of the study sites.
Two separate sample sets were collected – for the determination of bulk density, ash content and chemical parameters (pH,
concentrations of total carbon (TC), nitrogen (TN), phosphorus (P), potassium (K), calcium (Ca), and magnesium (Mg)). The
samples were collected with a volumetric 100 $cm^3$ cylinder (Cools and De Vos, 2010) at 10 cm intervals to a depth of 50 cm.
Two additional samples were taken from soil depths of 50-75 and 75-cm with a soil auger. Soil samples collected for
determination of bulk density were oven-dried (105 °C) and weighed (LVS ISO 11272:2017), while soil samples for chemical
analyses were prepared by air drying (≤40 °C), sieving and homogenizing (LVS ISO 11464:2006). Organic carbon (Corg)
content was calculated by multiplying the ash content measurement result derived soil organic matter content by factor 0.5,
thus assuming that organic matter is 50% Corg (Pribyl, 2010). Once per month, soil water samples were collected from separate
piezometers (7.5 cm in diameter) explicitly installed for water chemical analysis, not the ones used for WTL measurements.
Water chemical parameters such as water pH, electrical conductivity (EC), and concentrations of dissolved organic carbon
(DOC), total nitrogen (N), nitrate ($NO_3^-$), ammonium ($NH_4^+$), and phosphate ($PO_4^{3-}$) ions were determined.
All soil and water analyses were done in an ISO 17025 certified laboratory using ISO standard methods (Table S 3).

**1.5 Biomass and litter measurements**


Dry matter biomass of the total annual foliar fine litter (fLF) and coarse woody litter (cLF), ground vegetation of herbaceous
(vascular) plants (GV), moss production (MP), and tree fine-root (FR) production (FRP) were determined, and their C contents
analyzed in all study sites. Biomass of dwarf shrubs, moss and total belowground biomass was measured in some sites only
(Table S 8). fLF and cLF samples were collected once every four weeks, and GV, moss, FRP, dwarf shrubs and total root
biomass samples were collected once during the entire study period.
To avoid double accounting of foliar litter biomass, all fine fractions of litter and branches with a diameter of up to 1 cm and
a length of up to 10 cm were considered fLF. Branches with larger dimensions were considered cLF. fLF biomass samples
were collected with conical litter traps (area 0.5 m$^2$) set one meter above the ground (Latvia) or with square mesh frames (0.5
x 0.5 m) placed on the ground (Estonia and Lithuania). In each study site, five replicate litter collectors were placed in the
centermost subplot of the transect.
GV aboveground (aGV) and belowground (bGV) biomass and MP biomass samples were collected in 2021 in five replicates
per subplot. GV biomass was collected from square sampling points with an area of 0.0625 cm$^2$. GV belowground biomass
was collected from the top 20-30 cm of the soil layer. In the process of biomass determination bGV biomass was separated
from tree roots by wet sieving and morphological properties. MP biomass samples were collected by anchoring a square mesh
(0.01 m$^2$) on the moss at the end of vegetation season and collecting the moss biomass that grew through the mesh during the
next growing season. Also, GV samples were collected at the end of the growing season.
To estimate FRP, a modified ingrowth core method (Laiho et al., 2014) was applied. The method is based on a cylindrical
mesh bag (diameter 2.5 cm, mesh size 2 mm) filled with peat collected from the subplot. Ingrowth cores were installed in each
subplot in five replicates and removed from the soil after two growing seasons. In addition, total root biomass was estimated
by collecting undisturbed sample cores (18 cm$^2$) from the 0-40 cm soil layer. The collected samples were transported to the
laboratory, where the biomass of the ingrown FR was determined by morphological properties after wet sieving and separating
GV roots from tree roots.
All biomass samples were oven-dried (70 °C), weighed and milled prior to further analysis. Chemical analyses were performed
according to ISO standard methods (Table S 3).

**1.6 Estimation of annual soil and forest floor carbon balance**


We estimated the annual soil and forest floor (including ground vegetation) C balance of the sites by combining C input and
output: either Rtot or Rhet were used to represent the output of C, while the C inputs by plant litter were used identically for
both approaches. Consequently, results acquired by using Rhet as C output represent soil C balance, while approach with Rtot
– the C balance of forest floor. While we directly measured Rhet, we utilized the Rhet value derived from the results of Rtot.
Such an approach was necessary because our Rhet values were consistently higher than Rtot in numerous study sites (Figure
S 4). This appears to be an artifact, which explains our decision not to use directly measured Rhet (see Results and Discussion).
It made more sense to use Rtot because relying on Rhet would overestimate soil C loss. Rhet, which excludes autotrophic
respiration, unlike Rtot, should not be higher than Rtot.
To estimate annual C output, the results of the instantaneous Rtot measurements were first interpolated to annual cumulative
Rtot during the study period. Interpolation was carried out by evaluating the relationship between Rtot and soil temperature
measured in each study site and constructing site-specific regression equations for the purpose. Hourly Rtot were then
calculated using the hourly soil temperatures collected by data loggers at each study site. Consecutively, annual Rtot was
calculated by summing the interpolated hourly emissions in a specific study year.
We derived annual Rhet from estimated annual Rtot empirically using equation (2). The equation characterizes the relationship
between soil surface respiration (Rs) and Rhet, it was created using results of previous studies (Jian, J. et al., 2021) in boreal
zone (Figure S 5). We assumed Rtot is equal with Rs, i.e., aboveground autotrophic respiration has a minor role in Rtot



(Hermans et al., 2022; Munir et al., 2017) and applied the equation to annual Rtot directly. Such assumption was justified by
observation that there was no relationship found between share of Rhet and Rs (p=0.14) in partitioned Rtot data analyzed.

$$Rhet = -0.7 + 0.78 \times Rs \qquad (2)$$

To estimate the annual C input, the measured annual litter biomass (Table S 7) was recalculated to C amount using biomass C
content values evaluated in the study (Table2). For the estimation of C balance, the annual C input during the study period was
considered to consist of fLF, aGV and bGV litter and FR litter (estimated based on FRP). Only these sources of C input were
used because their decomposition resulting $CO_2$ emissions are directly accounted for in the Rtot. We assumed that FR biomass
was essentially not changing over the study years, and thus we could assume that FRP equaled litter production. Since the root
ingrowth cores were removed from the soil after two growing seasons, the FRP estimate was calculated by dividing the FR
biomass in the cores by two (Bhuiyan et al., 2017). We also assumed that measured GV is equal to annual GV litter.

**Table 2: Mean C and N content (mean±SD) in dry matter of biomass (%).** Abbreviations: aGV and bGV – above- and belowground
biomass of herbaceous vegetation, FR – tree fine roots, M – moss, fLF – fine litterfall, cLF – coarse woody litterfall.

| Element | aGV | bGV | FR | M | fLF | cLF |
|---------|-----|-----|-----|-----|-----|-----|
| C | 49.34±2.45 | 50.95±2.02 | 51.21±5.16 | 48.38±2.13 | 52.50±0.25 | 53.88±0.67 |
| N | 2.18±0.64 | 1.53±0.43 | 1.47±0.44 | 1.10+0.75 | 1.30±0.41 | 1.04±0.20 |


The results on cLF, moss, dwarf shrubs at total root biomass (Table S 8) presented in the 'Annual litter and biomass production'
section were not factored into the C balance estimation. The values provided there are for informational purposes only (refer
to the Results and Discussion section for more details). The inclusion of cLF in C inputs would lead to biased C balance
estimation, as cLF cannot be representatively included in Rtot measurements. The reason for the exclusion of MP was that we
could not equate moss production directly to litter production, given that moss cover was not measured. Similarly, while we
measured the total biomass of roots and shrubs, the litter of those C pools was not estimated; hence data on shrubs and total
root biomass was also not applied in soil C balance estimation and the corresponding results should be regarded as descriptive
of the sites.
To summarize, soil C balance or forest floor C balance was calculated by summing either annual Rhet or Rtot, respectively
with C content of annual fLF, aGV and bGV litter and FR litter. The impact of drainage on C balance was assessed by
subtracting the estimated C balance in drained sites from undrained sites, utilizing C balance results obtained with both
approaches.
**1.7 Statistical analysis**
Statistical analyses were performed using the software R version 4.3.1 (packages 'MASS', 'stats', 'nlme', 'Hmisc',
'lmerTest'), using p=0.05 as the limit for statistical significance. The compliance of the data with the normal distribution was
checked formally with the Shapiro-Wilk normality test and visually by density and quantile-quantile (Q-Q) plots. Data on
instantaneous and annualized Rtot, WTL measurements and soil properties analysis results grouped by subcategories (drainage
status, dominant tree species, country) were compared for differences by pairwise Wilcoxon rank sum test with continuity
correction, and the p-values were adjusted by Bonferroni correction. Multivariate data relationships were observed through
Principal Component Analysis (PCA), and $CO_2$ emission-related relationships were confirmed by fitting raw data to the non-
linear Arrhenius equation (Lloyd and Taylor, 1994) and transformed data to linear mixed-effects models using the study site
as a random effect. As data transformation has a considerable impact on the results of statistical analyses, to improve the
normality of the data, a logarithmic and Box-Cox transformation was evaluated (Box and Cox, 1964; Liaw et al., 2021; Wutzler
et al., 2020). A method that achieved the best conformity to the normal distribution was used to transform the data. The





performance of elaborated models for flux data interpolation by continuous soil temperature measurements was compared by
root mean square error of prediction (RMSE). Figures are prepared by using packages 'ggplot2', 'corplot', 'ggbiplot'.
A descriptive evaluation of the hypothesis was evaluated by segregating instantaneous and annualized Rtot data by country
origin and checking for differences by PCA and by pairwise Wilcoxon rank sum test. Formal testing of the hypothesis was
performed by evaluating the significance of the country variable impact on the relationship between soil temperature and Rtot.
The exclusion of litter data from hypothesis testing was justified by litter being a proxy of Rtot.
**2 Results**
**2.1 Soil and soil water characteristics**
The peat layer depth in the study sites with drained soil ranged from 27 to 212 cm (mean 81±47 cm) and in undrained sites
from 100 to 230 cm (mean 167±49 cm). Soil bulk density (0-30 cm depth) in the drained sites (mean 314±215 kg m$^{-3}$) was
characterized by both higher variation and higher mean density (p=0.003) compared to undrained sites (mean 168±32 kg m$^{-3}$).
Soil drainage status had no impact on Corg content (p=0.11, total mean 416±130 g kg$^{-1}$). However, drained soils had a higher
mean C:N ratio (22±7; p=0.01) than the undrained soils (17±3) (Table S 4). A trend could be observed that undrained soils
had higher nutrient concentrations and higher pH than the drained soils (Figure 2).

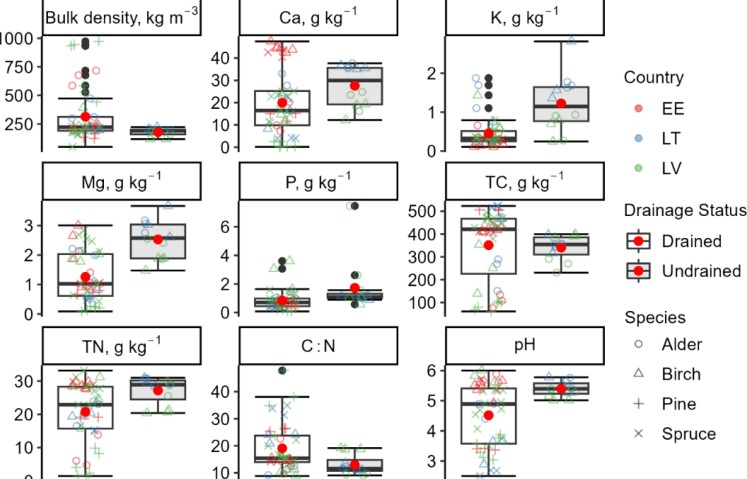


**Figure 2: Variation of soil chemical and physical properties at soil depth 0-30 cm.** The clear box represents the drained, grey-shaded
box the undrained sites. The bottom and top edges of the box represent the 25$^{th}$ and 75$^{th}$ percentiles, summarizing the interquartile range
(IQR). The whiskers extend to the smallest and largest values within 1.5 × IQR from the 25$^{th}$ and 75$^{th}$ percentiles, respectively. Black dots
mark outliers. A red dot and a solid horizontal line in the box indicate mean and median values, respectively.
The range of mean WTL was from −23 to −112 cm (mean −60±25 cm) in the drained sites and from −7 to −17 cm (mean
−13±4 cm) in the undrained sites, respectively. In the undrained sites, the WTL was mainly rather elevated (see interquartile
range in Figure 3) and had comparably smaller variation (mean standard deviation 16 cm) than in the drained sites (mean
standard deviation 23 cm); however, in all sites except LTC108, WTLs below 30 cm were also observed (Figure 4). In the
undrained sites, the range of min-max WTL was from 3±3 cm to −63±27 cm, while the WTL in drained sites had a greater
absolute variation and ranged from −14±19 cm to −104±28 cm.





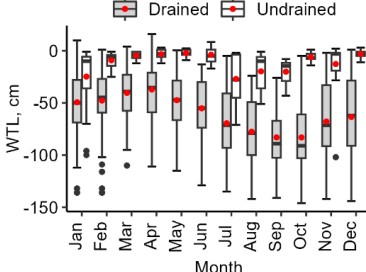

**Figure 3: Yearly variation of water table level (WTL) in the study sites.** The edges of the box represent the 25th and 75th percentiles, encapsulating the interquartile range (IQR). The whiskers extend to the smallest and largest values within 1.5 * IQR from the 25th and 75th percentiles, respectively. Black dots mark outliers. A red dot and a solid horizontal line indicate the average values of the date represented – mean and median, respectively.

The concentrations of all measured chemical parameters in the soil water, except for $NH_4^+$, were, on average, higher (p<0.05) in the drained sites (

Figure S 1). The most remarkable differences in mean concentrations were observed in the DOC, N, and $NO_3^-$ concentrations, which in the water of undrained sites were, on average, 1.5, 3.2, and 10 times higher, respectively (Table S 5).

## 2.2 Instantaneous total respiration

In the drained sites, the mean instantaneous Rtot varied from 48 to 125 mg $CO_2$-C $m^{-2}$ $h^{-1}$ and from 38 to 80 mg $CO_2$-C $m^{-2}$ $h^{-1}$ in the undrained sites (Figure 4: 44). The relative variations of the instantaneous Rtot in drained (CV=90±9%) and undrained (CV=106±29%) sites were comparable. Although the study sites represented a broad soil WTL gradient, no significant impact of the site mean WTL on the mean instantaneous Rtot emission was observed (r=0.16, p>0.05). Furthermore, no significant correlations were found between instantaneous Rtot and groundwater parameters.

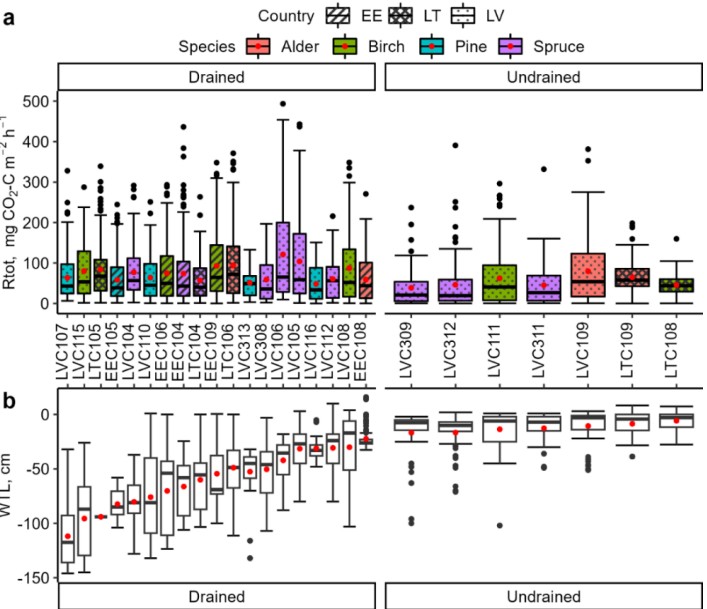

**Figure 4: 4 Variation of instantaneous total respiration (Rtot, panel "a") and water table level (WTL, panel "b") in the study sites.** The clear box represents the drained, grey-shaded box the undrained sites. The bottom and top edges of the box represent the 25th and 75th percentiles, summarizing the interquartile range (IQR). The whiskers extend to the smallest and largest values within 1.5 × IQR from the 25th and 75th percentiles, respectively. Black dots mark outliers. A red dot and a solid horizontal line in the box indicate mean and median values, respectively.



Mean Rtot in sites with the same drainage status did not differ (p>0.05) between countries (Figure S 2). Stratified by country,
the instantaneous Rtot in drained sites (mean: $76\pm3$ mg $CO_2$ C m$^{-2}$ h$^{-1}$) was overall higher (p<0.05) than those from undrained
soil (mean: $56\pm5$ mg $CO_2$-C m$^{-2}$ h$^{-1}$). The measured Rtot of undrained soil were smaller in both Latvia (mean $57\pm6$
mg $CO_2$-C m$^{-2}$ h$^{-1}$) and Lithuania ($55\pm6$ mg $CO_2$-C m$^{-2}$ h$^{-1}$) compared to Rtot from drained soil in the Baltic states ranging
from mean $72\pm4$ to $79\pm5$ mg $CO_2$-C m$^{-2}$ h$^{-1}$ (Figure 5:5, panel "a").


**Figure 5:5 Mean instantaneous total respiration (Rtot) by drainage status and country (a) or dominant tree species (b).** Error bars
indicate confidence interval. Shared letter indicates that differences are not significant.

There were few apparent differences in the mean Rtot between stands of different tree species (Figure 5:5, panel "b"). Rtot
was lowest at undrained sites dominated by spruce and highest at drained sites dominated by birch. Furthermore, Rtot values
in drained birch-dominated sites were not significantly different from those in both drained spruce- and undrained alder-
dominated sites. Rtot was significantly different (p<0.05) between coniferous forest sites with different dominant tree species
and soil moisture regimes, where Rtot ranged from mean $42\pm7$ mg $CO_2$-C m$^{-2}$ h$^{-1}$ in undrained spruce forests to $59\pm4$ and
$81\pm6$ mg $CO_2$-C m$^{-2}$ h$^{-1}$ in drained pine and spruce forests, respectively. In deciduous stands, the moisture regime and
dominant tree species had less impact on the mean flux; Rtot was higher (p<0.05) in drained birch stands (mean $84\pm5$
mg $CO_2$-C m$^{-2}$ h$^{-1}$) than those in undrained sites ($56\pm8$ mg $CO_2$-C m$^{-2}$ h$^{-1}$), while in alder stands the mean Rtot was similar
regardless of the soil moisture regime (total average $67\pm9$ mg $CO_2$-C m$^{-2}$ h$^{-1}$), (Figure 5:5, panel "b"). In drained coniferous
and deciduous sites, the mean Rtot was similar, but in undrained sites, emissions in deciduous forests were about 40% higher.
Evaluating the impact of country, drainage status, dominant tree species, WTL, and WTL category (above or below 30 cm) on
the relationships in mixed-effects models predicting Rtot by soil temperature, it was observed that all WTL-related model
factors had a significant impact, but the country and dominant tree species had no role in Rtot prediction. The impact of
drainage is also indicated by the mean measured Rtot, which was $87\pm3$ mg $CO_2$C m$^{-2}$ h$^{-1}$ in drained and
$57\pm3$ mg $CO_2$C m$^{-2}$ h$^{-1}$ in undrained sites if WTL depth of 30 cm threshold was considered as a threshold separating drained
and undrained soil. However, including WTL-related factors did not improve the fit of models (Table S 9) and prediction
improvement was negligible. These results confirm that neither country nor dominant tree species significantly impact
instantaneous Rtot.
**2.3 Annual total respiration**
The strongest correlation between instantaneous Rtot and soil temperatures measured at different depths was found for soil
temperature at 10 cm depth, with a mean Pearson correlation coefficient (r) of $0.86\pm0.04$ across the study sites. For the other
soil depths (5, 20, 30, 40 cm), r ranged from $0.71\pm0.07$ to $0.79\pm0.05$. Accordingly, soil temperature at 10 cm depth (Figure S
3) was used in constructing Rtot prediction models and for emission interpolation. Linear models developed using Box-Cox
transformed data provided the best Rtot prediction power. A lambda value of 0.3411 was used for all data transformations, as
individual data transformations for each site resulted in comparatively less successful data normalization. With this approach,
the RMSEP (Root Mean Square Error of Prediction) of instantaneous Rtot predictions for individual sites decreased by an

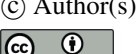



average of 16±14%, compared to linear models with logarithmically transformed data or non-linear models with untransformed
data (Table S 6). Interestingly, while the R10 (forest floor respiration at 10 C°) value increased by 20±7%, the estimated
cumulative annual Rtot decreased by 2±9%.
Annualized Rtot indicated similar mutual relationships among the study site dominant tree species and drainage status
categories as the instantaneous Rtot. Consequently, the estimated annual emissions from drained sites among the Baltic states
did not differ significantly (overall mean 6.21±0.43 t $CO_2$-C ha$^{-1}$ year$^{-1}$) and were generally somewhat higher than Rtot from
undrained soils in Latvia and Lithuania (Figure 6, panel "a"). Also, in undrained soil category, no significant difference was
found between the countries (total mean 4.38±1.20 t $CO_2$-C ha$^{-1}$ year$^{-1}$).

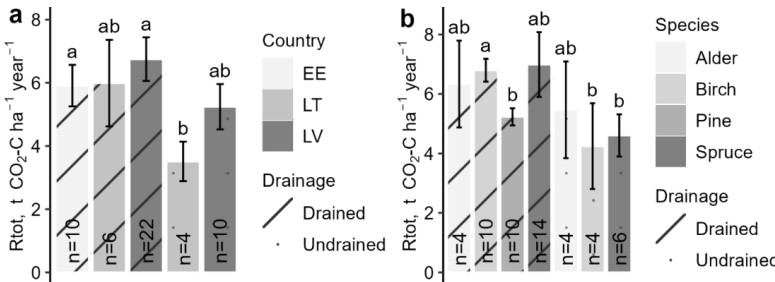

**Figure 6: Annualized total respiration (Rtot) in study sites stratified by drainage status and country (panel "a") or dominant tree species (panel "b").** Error bars indicate confidence interval. Shared letter indicates that differences are not significant.

Similarly, when categorizing data according to drainage status and dominant tree species, the differences between categories
in the annualized Rtot are statistically less significant than in the case of instantaneous Rtot data Figure 6, panel "b"). For
instance, the annual Rtot, regardless of the soil drainage status, did not significantly differ in most forests with various dominant
tree species. Among the drained sites, the lowest mean annual Rtot was estimated for pine forests
(5.23±0.29 t $CO_2$-C ha$^{-1}$ year$^{-1}$), while in spruce, birch, and alder forests, the means were similar (p>0.05), amounting
6.71±0.31 t $CO_2$-C ha$^{-1}$ year$^{-1}$. Emissions from undrained soils in alder, birch, and spruce forests are lowered, ranging from
4.6±0.71 in spruce forests to 5.47±1.63 t $CO_2$-C ha$^{-1}$ year$^{-1}$ in alder forests (overall mean 4.86±0.71).
The correlation between Rtot and WTL was low, however, the drainage status (drainage dich presence) impact on Rtot is
indicated by the PCA results, where undrained sites tend to have more similar characteristics, i.e., higher comparability.
Meantime drained sites show greater diversity when both instantaneous and annualized Rtot data is evaluated. However, clear
patterns of dominant tree species and country impact on Rtot are not recognized by PCA (Figure S 7 and Figure S 8).
When comparing the chemical and physical properties of different soil layers with the estimated annual cumulative Rtot, as
well as the measured mean instantaneous Rhet and Rtot, mean measured Rhet consistently shows a higher correlation with all
evaluated soil parameters. The only exception is Corg, where in all correlation combinations, it was not present (r around -
0.1). Excluding Corg, the other soil chemical parameters generally have a low to moderate correlation (mean r=0.4) with Rhet.
The highest correlation is with pH, K, Mg, and P (mean r=0.5±0.07, p<0.05), and it is consistent across all evaluated soil
layers, while correlation with BD (mean r=-0.2, p>0.05) tends to increase with deeper soil layers reaching the highest
correlation (r=-0.3) in layer 20-30 cm (Figure S 6). In addition, higher C:N ratio is associated with lower Rhet emissions
(mean r=-0.4, p<0.05).

### 2.4 Annual litter and biomass production

The estimated mean biomass of different plant litter categories in both drained and undrained sites were mostly similar,
typically not differing by more than 20%. Only fLF and FRP tended to be considerably higher in the drained sites, FRP on
average even more than twice as high. Compared to undrained sites, bGV in drained sites was about 20% higher on average,




while aGV was about 20% lower on average (Figure 7). However, regardless of the soil drainage status, the proportion of aGV
in the total GV biomass was 54±18%. The estimated moss biomass dry matter (dm.) averaged 5.02±0.87 t dm. ha⁻¹, and MP
averaged 0.98±0.25 t dm. ha⁻¹, or 22±10% of the total moss biomass (Table 3). In total, the sum of annual forest floor biomass
production (excluding small shrubs), cLF and fLF in the drained and undrained study sites was 9.68±2.95 and 8.68±2.10 t dm.
ha⁻¹, respectively.

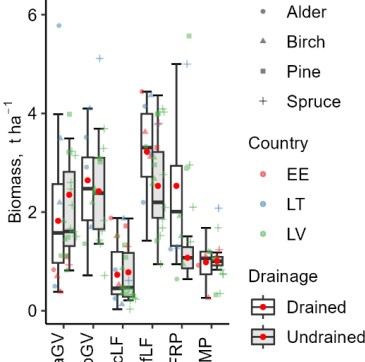


**Figure 7: Variation of biomass measurement results.** Abbreviations: aGV and bGV – above- and belowground biomass of herbaceous
vegetation, cLF – coarse woody litterfall, fLF – fine litterfall, FRP – tree fine root production, MP – moss production (assuming 100%
projection). The bottom and top edges of the box represent the 25th and 75th percentiles, summarizing the interquartile range (IQR). The
whiskers extend to the smallest and largest values within 1.5 × IQR from the 25th and 75th percentiles, respectively. Black dots mark outliers.
A red dot and a solid horizontal line in the box indicate mean and median values, respectively.
**Table 3: Biomass (mean±CI, t dm. ha⁻¹) measurement results stratified by drainage status.** Abbreviations: aGV and bGV – above- and
belowground biomass of herbaceous vegetation; respectively; S – small shrubs; FRP – tree fine root production; M – moss; MP – moss
production (assuming 100% projection); fLF – fine litterfall; cLF – coarse woody litterfall; RB - total root biomass.

| Category | Drained | Undrained |
|----------|---------|-----------|
| aGV | 1.82±0.52 | 2.35±1.61 |
| bGV | 2.89±0.85 | 2.42±0.84 |
| S | 0.84±0.45 | 4.27±2.5 |
| FRP | 2.53±0.77 | 1.08±0.57 |
| M | 5.02±0.87 | - |
| MP | 0.98±0.25 | 1.01±0.23 |
| fLF | 3.22±0.44 | 2.53±1.06 |
| cLF | 0.73±0.27 | 0.78±0.62 |
| RB | 39.3±11.1 | 52.7±18.7 |


Both bGV (r=|0.6|) and FRP (r=|0.7|) biomass have a significant negative correlation with soil pH but a positive with the C:N
ratio in soil layer 0-30 cm. Additionally, FRP has a significant negative correlation (r=-0.7) with the content of biogenic
elements (N, Ca, Mg) in the soil. No explanatory factors for aGV and MP biomass have been identified. Moderate correlation
(r=0.5, p<0.05) is found between stand age and fLF, while cLF has a weak relationship with stand parameters.
The study's estimated sum of annual gross ground vegetation biomass production and litter in sites with drained soil ranges
from 4.58 to 8.29 t C ha⁻¹ year⁻¹ (mean 5.91±1.10 t C ha⁻¹ year⁻¹), while in sites with undrained soil, it ranges from 3.41 to
5.40 t C ha⁻¹ year⁻¹ (mean 4.72±1.16 t C ha⁻¹ year⁻¹) (Figure 8).



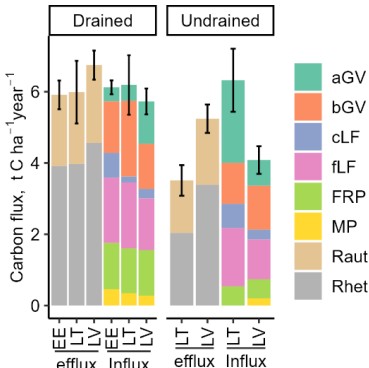

**Figure 8: Forest floor, including ground vegetation, carbon balance (sum±combined CI).** Abbreviations: aGV and bGV – above- and belowground biomass of herbaceous vegetation; respectively; cLF – coarse woody litterfall; fLF – fine litterfall; FRP – tree fine root production; MP – moss production (assuming 100% projection); Raut – soil autotrophic respiration; Rhet – soil heterotrophic respiration. The assumption is used that biomass is equivalent to the average results obtained from sites with the same drainage status in other countries in cases the corresponding carbon pool was not estimated in some countries, such as bGV and FRP in EE and FRP in certain sites in LT and LV. Figure should not be interpreted as soil C balance (see Materials and Methods).

**2.5 Annual soil and forest floor carbon balance**

Based on assumption (2) the estimated Rhet proportion of Rtot varied between 54 and 71% (mean 65%). Consequently, the estimated annual gross C losses from drained soil in the form of Rhet emissions in the study sites ranged from 2.36 to 7.49 t $CO_2$-C ha$^{-1}$ year$^{-1}$ (mean 4.30±1.20 t $CO_2$-C ha$^{-1}$ year$^{-1}$), while for undrained soils, gross C loss range from 1.63 to 4.68 t $CO_2$-C ha$^{-1}$ year$^{-1}$ (mean 3.00±0.99 t $CO_2$-C ha$^{-1}$ year$^{-1}$). In drained and undrained sites, C input applied in C balance estimation ranged from 3.81 to 7.03 t C ha$^{-1}$ year$^{-1}$ (mean 5.20±0.91 t C ha$^{-1}$ year$^{-1}$) and 2.89 to 5.98 t C ha$^{-1}$ year$^{-1}$ (mean 4.19±1.10 t C ha$^{-1}$ year$^{-1}$), respectively. When these averages are compared with the estimated Rhet C losses for both drained and undrained soils, it is found that the soil C stock did not diminish during the study period, irrespective of drainage status. For instance, direct comparison of the mean estimated C influx and efflux in drained and undrained soils shows that during the study period soil C stock increased by mean 0.9±1.51 and 1.19±1.48 t C ha$^{-1}$ year$^{-1}$, respectively. These results show that drainage of soil reduces C sequestration capacity by average 0.29 t C ha$^{-1}$ year$^{-1}$. However, a more conservative C balance estimation approach, assuming C efflux is equal to measured Rtot, estimated that annual net C efflux of forest floor in drained sites was 1.05±0.98 t C ha$^{-1}$ year$^{-1}$ on average, whereas in undrained sites a C source amounting mean 0.48±1.61 t C ha$^{-1}$ year$^{-1}$ (Table S 10). This approach reveals that forest floor in drained sites contributed to a higher net C efflux, averaging 0.57 t C ha$^{-1}$ year$^{-1}$. Consequently, both C balance estimation approaches consistently indicate a negative impact of drainage - 0.43±2.69 t C ha$^{-1}$ year$^{-1}$.

**3 Discussion**

The study highlights the critical need to assess the accuracy of Rhet measurements before their application in subsequent analyses and result compilation. In our case, we observed errors in the measurements because it was possible to compare the results with Rtot. In studies where Rtot measurement is not the primary objective, control measurements near a trenched area could be introduced as a necessary Rhet measurement quality assurance measure. This issue led us to use Rtot as a proxy to characterize soil emissions, which likely introduces additional complications in assessing the impact of influencing factors on soil C balance. Therefore, in the subsequent sections, we will explore the relationships identified in connection with individual C balance components.





**3.1 Soil heterotrophic respiration**

The higher correlation of all soil parameters except Corg with Rhet is likely due to higher role of autotrophic respiration component in Rtot emissions which introduces additional "noise". While no correlation was found between flux and Corg content of the soil, it is probably because the soil in the study sites does not represent a wide gradient of Corg concentration. It is observed that higher Corg content is related to lower soil $CO_2$ emissions, as in peatlands lower soil C content is often related to higher nutrient availability (Jauhiainen et al., 2023). Although uncorrelated Corg and Rhet observations are common. Probably our study had limited opportunities to identify relationships because all the study sites are classified as nutrient rich. Negative correlation found between C:N and Rhet allows to assume that if C:N ratio is related to the degree of soil mineralization, it is more indicative of past elevated emissions rather than currently increased mineralization process. In other words, increased emissions likely occurred when the soil was recently drained compared to current situation when ecosystem likely tends towards new equilibrium after the initial disturbance. Despite the scarcity of research on the long-term impact of drainage on soil $CO_2$ emissions, some evidence suggests that the role of Rhet in Rtot declines in time since drainage (Munir et al., 2017), and effects of initial disturbances can be mitigated after 100 years (Hommeltenberg et al., 2014; Vanguelova et al., 2019). This might explain the absence of significantly increased emissions in the historically drained peatlands that we investigated. Another related reason emissions from drained soils may not be significantly increased is soil compaction resulting from the drainage process. This is indicated by the increased BD of drained soil, which is associated with lower soil Rhet emissions (mean r=-0.2). The reason may be reduced soil porosity limiting gas exchange between the soil and the atmosphere (Ball, 2013; Novara et al., 2012). Reduced porosity also leads to higher water level fluctuations in drained peat soils (Figure 3 and Figure 4: 4), consequently stimulating redox processes and $CO_2$ emissions (Wang et al., 2018). In the case of BD correlation, a different trend is observed compared to other soil parameters: instead of being comparably uniform across soil layer it tends to increase with deeper soil layers, being highest (r=-0.3) in the 20-30 cm layer, which is also the layer that showed the highest correlation between Rhet and chemical properties. Thus, while this layer may be prone to a high decomposition rate, flux driving processes seem to be countered by increased soil compaction. Therefore, a long time after drainage, although the soil $CO_2$ flux from drained soil is likely higher ($4.30\pm1.20$ t $CO_2$-C ha$^{-1}$ year$^{-1}$) they may not differ significantly from undrained soil (mean $3.00\pm0.99$ t $CO_2$-C ha$^{-1}$ year$^{-1}$) as shown by Rtot derived Rhet estimation of this study.

**3.2 Soil heterotrophic respiration interpretation issues**

Acquired Rhet data was excluded from annual soil C balance estimation as there was sufficient evidence of error in the acquired results. One of the probable main reasons for the failure of Rhet measurements is indicated by the correlation (r) achieved between soil temperature and Rhet, ranging from a mean of $0.28\pm0.12$ to $0.51\pm0.12$. Thus, significantly lower compared to correlation found with Rtot, indicating measurement error as both root and microbial respiration are temperature dependent (Davidson and Janssens, 2006). In our case reduced correlation seems introduced generally due to high emission outliers at elevated soil temperatures, which may lead to considerable Rhet overestimation by flux interpolation models restricted to predict reduced emissions at increased temperatures when soil moisture regime does not favor microbial activity (Khomik et al., 2009; Yueqian and Sc, n.d.). The reason for affected Rhet measurement quality may have been the study design's deficiency in measuring soil temperature. Temperature readings, taken at the center of the subplot, may not have accurately reflected potential temperature differences between areas with intact vegetation and trenched sections. The environmental parameter measurements did not consider that soil temperature and moisture conditions in trenched areas might differ. The error of measured Rhet is evident in the observed relationship between Rhet or Rtot and temperature, indicating higher Rhet emissions at the same temperatures (Figure S 4).

Trenching altered soil conditions (Ojanen et al., 2012) can be a reason for biased Rhet measurements (Pumpanen et al., 2010) as soil respiration is influenced not only by soil temperature but also by water availability (Davidson and Janssens, 2006). Not accounting of moisture regime in interpolation of flux measurement results can lead to overestimation as Rhet prediction



models have to be available to predict lower emissions at even increased temperatures if soil moisture is limiting microbial
activity (Jovani-Sancho et al., 2018; Liaw et al., 2021). No availability of temperature and soil moisture continuous and manual
measurements directly in trenched spots did not allow to empirically address these issues.
Also, direct comparison of measured Rtot and Rhet, ignoring temperature relationships, indicates that comparability has been
disturbed as Rhet exceeded Rtot measurement results. Impact of analytical instruments used was excluded as bias of
measurement results was not observed during initial comparison of two used instrumental methods (gas chromatograph and
portable analyzer) performed in controlled conditions. However, it must be noted that during this comparison the same chamber
was used. Therefore, probably reasons for Rhet measurement errors may be not only soil conditions altered by trenching, but
also: disturbance of natural conditions may have been further stimulated by covering trenched area by geotextile between
measurements; differences in flux measurement technical aspects - chamber sizes, measurement time, application of fan for
mixing air inside the chamber headspace.
Even if the trenching process did not alter soil temperature and moisture levels, a significant source of error could stem from
the decomposition of severed roots. Soil trenching was conducted before winter, and by spring, when measurements began,
the cut roots decomposition accelerated and that reflected in Rhet measurements. To assess the potential impact of the cut
roots, we collected total belowground biomass samples from the top 40 cm of soil using a soil probe and found that the total
root biomass in drained and undrained was, on average, $39.3\pm11.1$ and $52.7\pm18.7$ t ha$^{-1}$, respectively. Considering that around
50% of roots can decompose over two years (Straková et al., 2012; Moore et al., 1999), the study period's underground biomass
decomposition could have led to a significant artificial increase in measured Rhet of drained (11.67 t $CO_2$-C ha$^{-1}$ year$^{-1}$) and
undrained (14.37 t $CO_2$-C ha$^{-1}$ year$^{-1}$) soils. Specifically, the decomposition of roots may have raised the Rhet value by 4.90
and 6.59 t $CO_2$-C ha$^{-1}$ year$^{-1}$, respectively. Although this estimation is rough, it quite well illustrates the potential
overestimation by root decomposition, especially since the measured Rhet in the study exceeded the Rtot by an average of
$5.8\pm3.1$ t $CO_2$-C ha$^{-1}$ year$^{-1}$. Therefore, the primary source of error in Rhet measurements was likely the decomposition of
roots.

### 3.3 Total respiration

The gradient of the mean air temperature from Estonia to Lithuania varied from 6.4 to 8.5 °C. However, no significant
differences in Rtot measurements were observed across the countries. Similarly, no clear impact of dominant tree species on
Rtot was found. Mean instantaneous Rtot measurements results indicate a greater relationship with the dominant tree species
is the study sites, compared to the results of annual cumulative Rtot estimated. However, in both cases, the influence of species
on Rtot is weak. Which was also confirmed by PCA and mixed-effects modeling. This points to difficulties of evaluating the
role of dominant tree species on emissions. However, there is some evidence found that emissions in undrained sites tend to
be higher in deciduous stands, particularly alder stands, according to results of measured instantaneous emissions. The
enhanced soil $CO_2$ efflux observed in the presence of alder can be attributed to the symbiotic nitrogen fixation process
associated with these trees (Warlo et al., 2019), which increases nitrogen availability in the soil. Nitrogen availability, in turn,
can stimulate decomposition processes, leading to a higher rate of $CO_2$ release. Although statistically unconfirmed, a tendency
can be noticed that in drained sites Rtot emissions tend to be higher in birch stands, but lower in pine forests. Also, previous
studies indicated that deciduous stands are responsible for higher $CO_2$ emissions (Jauhiainen et al., 2023).
While both drainage status and WTL threshold above or below 30 cm can be used as a predictor of Rtot, meaningful correlation
between WTL and Rtot was not found. Furthermore, although the absolute variation of the WTL was higher in drained sites,
the relative variation in both WTL level and Rtot was indifferent to the drainage status. The observation suggests that raised
WTL conditions in undrained sites, while decreases Rtot emissions, does not guarantee higher resilience to moisture regime
disturbances, i.e., more stable emissions. Main reason is just as the presence of drainage ditches cannot consistently lower
WTL both spatially and temporally, in undrained sites too, WTL frequently falls below 30 cm (Butlers et al., 2023) ensuring



aerobic conditions in soil layers containing labile organic matter. Furthermore, this typically happens in summer (Butlers et
al., 2023) when increased temperatures further promote organic matter mineralization of undrained soil.  Role of WTL
dynamics is reflected also in results in PCA, showing increased diversity of drained sites likely due to higher absolute variation
in WTL depth. This may be the reason complicates quantification of relationships between flux and the affecting factors,
especially in drained sites.
To aim towards accurate Rtot annualization using periodic flux measurements, data interpolation by modeling approaches is
necessary. Both advantages and shortcomings of different data transformation methods and modeling approaches are reported
by previous studies (Yueqian and Sc, n.d.; Wutzler et al., 2020; Liaw et al., 2021; Moulin et al., 2014; Box and Cox, 1964;
Khomik et al., 2009). Although the bias in predicted annual Rtot varied among study sites, the overall impact of different flux
modeling approaches on annual Rtot estimations was minimal. Specifically, the mean bias of results obtained through the
implementation of the Box-Cox transformation was -2±9%, indicating a rather consistent accuracy compared to other methods
used.

### 3.4 Carbon balance estimation

### 3.4.1 Carbon efflux

Since direct Rhet measurements were excluded from the soil C balance calculation, the C efflux (Rhet) was derived from Rtot
measurement results empirically using the Rhet/Rs factors of previous studies. Rhet and Rs values from database (Jian, J. et
al., 2021) on forests soil flux in the boreal zone were used, as existing experience suggests that organic soil emissions in
hemiboreal forests are more likely to align with boreal rather than temperate conditions (Krasnova et al., 2019; Heikkinen et
al., 2023; Bārdule et al., 2022; Butlers et al., 2022; Dubra et al., 2023; Lazdiņš et al., 2024). Choice of using only boreal data
tends towards use of higher share of Rhet, compared if temperate data were used, as illustrated in Figure S 4. This approach
aimed to avoiding the underestimation of soil C losses. The share acquired using boreal data is 0.65±0.04, while using
temperate data - 0,60±0,15, or around 10% difference. According to this approach we estimated mean Rhet of drained soil as
mean 4.30±1.20 t $CO_2$-C ha$^{-1}$ year$^{-1}$, which is slightly higher than mean Rhet of 3.71±0.53 t $CO_2$-C ha$^{-1}$ year$^{-1}$ found in
previous studies of forest organic soil (Jian, J. et al., 2021). However, conclusions or observations of studies in the boreal zone
may not be directly applicable to the hemiboreal zone. One reason for this is the larger removals by net ecosystem exchange
observed in the hemiboreal zone than northern forests, which also creates greater potential for C influx by litter to offset Rhet
C loss (Krasnova et al., 2019), which should be linked also to Rhet rates.
The role of ground vegetation autotrophic respiration in Rtot increases with its biomass (Munir et al., 2017). Consequently,
the applied approach of empirical Rhet calculation may have overestimated Rhet. This aspect is considerable in our assessment
of soil C balance. However, when estimating the impact of drainage on the soil C balance, any bias introduced was likely
negligible, because the mean ground vegetation biomass did not significantly differ between drained and undrained sites
(Δ=0.53 t dm. ha$^{-1}$). Consequently, any bias introduced in the calculation of Rhet is offset when making relative comparisons
of the soil C balance between drained and undrained sites to determine the impact of drainage.

### 3.4.2 Carbon influx

When interpreting the study results, it is essential to consider the C fluxes included in the C balance calculations. In estimation
of C influx, we considered data only for fLF, aGV, bGV, and FRP, excluding cLF, MP and dwarf shrubs. This approach was
chosen because it is rational to directly compare these measurements with Rtot, as the mineralization produced $CO_2$ emissions
of these litter is directly included in Rtot. However, including litter such as cLF, MP, and shrubs in the calculation would
overestimate C input into the soil, because cLF due to its dimensions and scarce coverage could not be objectively included in
chamber measurements. Furthermore, while fLF is relatively uniform in forest area, the coverage of mosses and dwarf shrubs
is not always so, therefore it is necessary to know their area of projection to be included in C balance estimation. One of the





solutions for incorporating cLF influx is to use assumptions on litter mineralization rate. One example of how the issue can be
tackled is the use of modeling approaches such as Yasso (Alm et al., 2023). Information on small shrubs and moss biomass
can be added in modelling as well by considering their annual production and turnover rate. For these reasons litter biomass
(mean $4.70\pm1.43$ t C ha$^{-1}$ year$^{-1}$) used in calculation of soil C balance likely overestimates soil C loss, as inclusion of cLF and
MP can increase soil C input by up to mean 0.9 t C ha$^{-1}$ year$^{-1}$, according to the study data.

### 3.4.3 Carbon balance

By applying different data aggregation approaches (e.g., by country or dominant tree species), varying results for soil C balance
estimation were achieved. For instance, by categorizing the data according to drainage status and dominant tree species and
adopting the approach of using Rtot as the C output value, we determined that forest floor, including ground vegetation, in
both drained and undrained sites were sources of $CO_2$ emissions. Specifically, in drained deciduous species sites, average
estimated net soil C efflux is $1.84\pm0.93$ t C ha$^{-1}$ year$^{-1}$, forest floor in coniferous stands show an estimated soil C sequestration
of $0.39\pm0.57$ t C ha$^{-1}$ year$^{-1}$. Meanwhile, forest floor in undrained deciduous stands experienced a mean C loss efflux $0.16\pm0.64$
t C ha$^{-1}$ year$^{-1}$, while in spruce stands showed a loss $0.9\pm1.46$ t C ha$^{-1}$ year$^{-1}$. Thus, confirming previous observations that
drained deciduous forests can be associated with higher soil $CO_2$ emissions (Jauhiainen et al., 2023) However, our estimates
of C balance by species may be influenced by the random effects associated with the study sites, as suggested by mixed-effects
modeling. Such data segregation approach is not unequivocally most appropriate if considering that the study did not find a
definite proof of country or dominant tree species impact on Rtot. Therefore, we should interpret these specific study results
with caution. We believe that estimating the C balance based on drainage status, without further stratification of results,
provided a more accurate assessment. Furthermore, it is evident that utilizing Rtot as the soil C output value leads to an
overestimation of soil carbon losses. Therefore, the focus should primarily be on analyzing the results of C balance estimation
by incorporating Rhet, assessing the capacity of litter to offset soil C losses. Additionally, it is crucial to examine the factors
influencing soil C influx and efflux to aim towards an accurate assessment of changes in soil C stocks of both drained and
undrained soils.
There are observations that soil $CO_2$ emissions are determined by soil nutrient status of the site (Meyer et al., 2013; Korkiakoski
et al., 2023), supporting assumption that nutrient-rich soils are likely a C source. However, such interpretation should be
exercised cautiously, considering complexity of forest floor C balance components and interactions. Some aspects noticed in
this study are a negative correlation found between nutrient availability and belowground biomass (bGV, FRP) confirming
previous observations that greater belowground biomass is associated with reduced nutrient availability (Zhang et al., 2024).
At the same time increased ground vegetation belowground biomass was associated with lower WTL. Which are two
countering effects in the study sites, as while WTL is increased in undrained sites, soil in these sites were nutrient richer. In
general, while higher organic matter decomposition rates can be expected for nutrient rich sites (Shahbaz et al., 2022; Hiraishi
et al., 2013a), also higher total soil C influx by litter can be expected by increased biomass growth, thus offsetting soil C loss
by Rhet. This is indicated by our study, as in both drained and undrained sites mean C amount of litter biomass exceeded
estimated Rhet.
Acquired empirical data segregated by drainage status indicated that both drained and undrained nutrient rich organic soil in
Baltic states is not a C source ensuring C removals of $0.9\pm1.51$ t C ha$^{-1}$ year$^{-1}$ and $1.19\pm1.48$ t C ha$^{-1}$ year$^{-1}$, respectively.
Such results can be found controversial by the general public, but results of studies showing soil as a C sink in afforested
peatlands is common (Minkkinen et al., 2018; Lohila et al., 2011; Bjarnadottir et al., 2021), preventing consensus that peatland
drainage is a measure associated with soil C stock loss. Preserved C stock is also indicated by similar mean estimated Corg
content in the top 30 cm of soil suggesting that the Corg stock in drained soils might not be at higher risk than undrained ones.
However, such an assessment, although providing indications, would not be correct for comparison because the C stock is
significantly influenced by the soil bulk density, which in 30 cm topsoil was on average 1.8 times greater in the drained soils



at the study sites, consequently almost doubling the C stock in the corresponding soil layer and meantime suggesting a soil compaction introduced by drainage. Noteworthy that the C:N ratio in drained soils is increased compared to undrained soils, however it can give a misleading impression of the degree of ongoing mineralization if nitrogen inputs and outputs from the soil are not considered (Ostrowska and Porębska, 2015). Nevertheless, assuming that the area was drained around a century ago, as was mostly done in this region (Zālītis, 2012), then according to the IPCC default emission factor of 2.6 t $CO_2$-C ha$^{-1}$ year$^{-1}$ (Hiraishi et al., 2013b), the Corg stock should have been already depleted in drained soils which is not the case. Drainage of peatlands does not necessarily result in a loss of soil C stocks (Minkkinen and Laine, 1998). Short-term soil C loss due to drainage induced increase in gross soil $CO_2$ emissions could already been offset by enhanced biomass growth (Hommeltenberg et al., 2014), as initial soil C stock can be restored after several forest rotations (Vanguelova et al., 2019). It is observed by a local soil inventory studies that C stocks of forestry drained peatlands are stable in nutrient-poor or moderate rich soil conditions, while there is evidence that nutrient-rich organic soil can lose C stock in the long term (Lazdiņš et al., 2024; Dubra et al., 2023). However, we did not find firm proof of that in this study.

## 4 Conclusions

The study indicates complex interactions between a range of factors, including water table level dynamics, soil compaction, availability of labile organic matter, and the litter sources and variation, determining the soil carbon balance. Thus, highlighting the importance of considering nutrient status and drainage status rather as proxies of the underlying factors influencing soil $CO_2$ fluxes than general predictors. While not confirmed with high certainty, indications have been observed that $CO_2$ emissions from soils in deciduous forests tend to be higher than in coniferous forests. However, estimated soil C influx and efflux did not conclusively demonstrate a loss of soil carbon stock within the study sites. The absence of a significant country impact on the estimated soil emissions suggests that a uniform approach for organic soil emissions estimation can be applied across the Baltic states.

Two approaches used for the carbon balance estimation provided contradictory results, with soil being estimated as a carbon sink regardless of drainage status, while the forest floor (including ground vegetation) was estimated to be a net source of $CO_2$ emissions. Likely a more accurate estimation would be to assume the carbon balance to be midway between these assessments. Consequently, during the study period, drained soils experienced a carbon loss of 0.07±1.80 t C ha$^{-1}$ year$^{-1}$, while carbon stocks of undrained soils increased by an average of 0.36±2.0 t C ha$^{-1}$ year$^{-1}$.

Despite the discrepancy in the evaluation of carbon balance in drained and undrained sites, the results regarding the impact of drainage on carbon balances were uniform. Both approaches showed a negative impact of drainage on carbon balance ranging between an average of 0.29 and 0.57 t C ha$^{-1}$ year$^{-1}$, with a mean of 0.43±2.69 t C ha$^{-1}$ year$^{-1}$.

## Data availability

Data used for carbon balance estimations is available at DOI: 10.5281/zenodo.11073425

## Author contributions

KS, RL, JJ, AL and KA developed a harmonized methodology. ABu, DČ, TS and MKS managed and processed the study data. ABu wrote the original manuscript, with JJ managing the writing process and incorporating insights from all team members, including significant reviewing contributions from RL. TS, ABā, IL, VS, HV, IL and AH provided critical reviews and edits to the manuscript.



**Acknowledgments**
The research is conducted within the framework of the project "Demonstration of climate change mitigation potential of
nutrients-rich organic soils in the Baltic States and Finland" (LIFE OrgBalt, LIFE18 CCM/LV/001158).
**Competing interests**
Authors have no competing interests to declare.

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
