# Peer review of "Soil and forest floor carbon balance in drained and undrained hemiboreal peatland forests"

_EGUsphere, 2024_

## Author Comment (AC1)

The manuscript by Butlers et al presents a data set of soil CO2 flux measurements obtained across 26 drained and undrained forested peatland sites in the Baltic states. Fluxes are measured using static chamber methods, and aim to inform the carbon balance according to main tree cover or geographical location (by country) between drained and undrained sites. Additional flux measurements on trenched subplots were intended to provide heterotrophic flux estimates to balance with C inputs, but these data are not used in the analysis.

I found the approach interesting, especially as it enabled a simultaneous analysis of drainage and tree species, which have been found to be linked to C losses in organic soils recently. The comparison by country was less engaging, especially as the main motivation was to define the Baltic region as a distinctive geoclimatic region that contrasts with boreal conditions.

We appreciate your efforts in providing feedback, and your constructive views and recommendations. Concerning the regional versus country-level analysis, the regional values are indeed the main outcome. However, we wish to add the transparency needed for potential use of the results in the national greenhouse gas inventories by also including the country-level results. The local conditions show a clear even if slight climatic gradient and thus it appears relevant to evaluate if this region should be further stratified in scope of emission factor elaboration or whether the region can be treated as uniform enough to apply uniform emission estimation.

Our initial goal was to use the measured heterotrophic respiration data to estimate the C outputs from soil. It is well known that trenching produces an "unnatural" amount of freshly dead root biomass, the decomposition of which is included in the heterotrophic respiration. We measured root biomass in the measurement plots that could be used to account for this "extra" C flux, and initiated a root decomposition experiment. Yet, it turned out that the heterotrophic respiration values were systematically very much higher than the total respiration values, and this could not be remedied by the auxiliary data available. The within-plot variation in root biomass was too high, and the available mass loss data not sufficient, to result in reliable values. Root biomass could not be sampled in the exact locations where the respiration was measured. In principle, such sampling could have been done after the gas flux measurements ended; that would, however, prevent continued flux measurements, which we wish to do to evaluate the duration of the strong root effect. Thus, we chose the more cumbersome, and certainly not perfect (as also discussed) option to use the total soil + ground vegetation respiration data instead. Similar approaches have been used before. Still, we chose to present and discuss also the data on heterotrophic respiration to add transparency, and to further inform the scientific community on potential bias introduced by the method. We are willing to further clarify these issues and our choices both in the Materials and Methods and Discussion sections if considered useful.

The manuscript is very long, and I struggled with the intended focus of it. Rather than building the study around testable hypotheses, it presents a wide range of measurements that are not all relevant to the main objectives. I think that a considerable evaluation of much sharper objectives and specific hypotheses is needed. This will guide the analysis which currently includes measurements that are not used for analysis of findings and results in a wide-ranging, similarly unfocussed discussion.

We very much sympathize with the struggles of the Reviewers. Their task is heavy, as unlike most readers, they are required to read through and thoroughly analyse the paper in full over a short time period. We are, naturally, willing to do our best to clarify the focus of the paper. However, we think that the wide range of measurements presented is a strong asset for the manuscript, and crucial for moving towards better understanding and quantification of forest soil carbon balance and the role of various sources of soil carbon inputs. Such increased understanding is the main purpose of the article, in addition more focused hypotheses are provided. It was recently pointed out (Jauhiainen 2019, 2023; a list of references is provided at the end of our responses) that papers reporting gas fluxes from drained organic forest soils generally fail to present site and environmental data, and/or methodological information, to the extent that it seriously hampers metaanalyses and syntheses that could lead to more accurate dynamic emission factors in the future. Therefore, we believe that the thorough description of the data and results, and the consequent discussion on the findings is necessary for this purpose and relevant for the article, as the details provided give the reader better understanding of specific conditions of the study sites on aspects previous studies are often lacking. We recognize that the article is lengthy, and to make it shorter we could, e.g., revise the approach of how information related to the Rhet measurementa are reported. We could leave the most relevant Rhet information in the article and move the further discussion of the issue to the supplementary materials, as the directly measured Rhet results were not used in estimation of the soil C balance. Such way, we would not lose the possibility to provide valuable information to interested readers while making the article shorter. Statistical testing was applied to evaluate aspects the article aimed to (L84-86), for this reason statistical differences of CO2 emission by countries or dominant tree species were performed. Approaches of descriptive and formal evaluation of hypotheses are described in L266-269. To enhance clarity, we can refine the definitions of the two hypotheses in the Introduction section as follows:

1. The first hypothesis states that annualized soil and forest floor respiration rates do not exhibit statistically significant differences among Estonia, Latvia, and Lithuania.
2. The second hypothesis posits that annualized soil and forest floor respiration rates vary significantly among sites dominated by different tree species.

Unfortunately, there were significant methodological issues that seriously hamper the interpretation of findings. Static chamber methods for CO2 flux measurements can be problematic, or require careful consideration of the concentration gradient over time. It is well described in the literature that a build-up of CO2 in the chamber space reduced diffusion-driven soil CO2 efflux, and these chambers were deployed for extended periods. A non-linear correction is hence likely to be needed, and there are numerous studies that describe methods to do so (see detailed comments below).

We beg to disagree with the assertion that the use of static chambers could have introduced flaws in our study. Prior to field measurements, we conducted comprehensive comparability tests of the methods employed (L482-485). These tests included an evaluation of the linearity of CO2 concentration increase within the chambers. Simultaneously, we measured CO2 concentration continuously using an EGM (employed in heterotrophic respiration measurements) and collected manual gas samples for gas chromatography, following the study procedure. Our analysis did not reveal any evidence of disrupted linearity, leading us to conclude that pressure build-up did not introduce bias. Noteworthily, the size of the chambers

used for our Rtot measurements was large (L122-123, a volume of 0.0655 m³ and an area of 0.196 m²), which evidently prevented the formation of nonlinearity.  Our team includes scientists with thorough experience in greenhouse gas flux measurements with chamber techniques, and procedures like checking the raw data for linear versus non-linear patterns over time are self-evident for us. We are also quite familiar with the literature dealing with these methodologies.

The static chamber method contrasts with an infrared analyser-=based chamber approach for flux measurements intrenched areas. These resulted in higher flux values than static chamber measurements, and are discarded on this basis alone. Small differences in temperature and moisture can not explain this discrepancy, and it is not clear why the IRGA based fluxes were deemed erroneous, and the static flux estimates assumed to be correct.

The decision to discard heterotrophic respiration measurements was based on multiple considerations beyond the observation of higher flux values compared to total (soil and forest floor) respiration (L436-500). These included the impact of trenching on soil temperature, indicated by a diminished correlation between temperature and flux (showing that trenching had affected soil temperature and likely also moisture regime), as well as the decomposition of severed roots, which could potentially increase the heterotrophic respiration by a factor of two judging by the measured root total biomass (L491-500). It was concluded that the overestimation of Rhet by root decomposition was the main source of error, as the belowground biomass at the sites was high. Consequently, the root trenching inevitably resulted in significant root decomposition emissions, critically affecting the quality of Rhet data and rendering these results unsuitable for use in C balance calculations. We performed several preliminary analyses and comparability tests of both methods (L482-485), which we did not explain in full in the manuscript. By these tests in controlled environment (soil emissions were measured in the laboratory using both sampling methods), it was concluded that the methods are comparable, further strengthening the conclusion of Rhet overestimation due to trenching. We can add more justification for our methodological choices if deemed useful, perhaps as Supplementary materials not to further extend the length of the manuscript.

I also missed a more critical engagement with flux results. Maximum values observed are very low for summer conditions, whilst minimum values observed in winter are very high. The constrained range of value across the season is unusual, and authors don't offer any suggestions as to why soil biological activity was maintained at considerable levels in the depth of winter with negligible photosynthetic C supply and likely frozen surface soils where most metabolic activity originates. I suggest that are careful re-analysis of flux estimates is needed, and a careful interrogation of flux responses with more than just temperature.

We can take a closer look at these in a revised version of the manuscript by, e.g., showing seasonality of soil emissions, and looking in more detail into winter emissions. Biological activity in winter is not related to air temperatures only, as, e.g., the emergence and thickness of the snow cover impact the soil temperatures.

The conclusions have to be much more carefully considered, given the considerable sources of uncertainty that the authors present quite openly. The data set may be suitable to derive total soil CO2 fluxes following a re-analysis, and estimating heterotrophic contributions using literature values may also be informative. But sweeping statements regarding source or sink

functions based on highly uncertain flux estimates and selective use of either boreal or temperate comparison values is not helpful.

In conclusion, I can not recommend the manuscript for publication in its present form. A slimmed down version with a clearer focus on testable hypotheses and a careful re-evaluation of flux estimates maybe worth considering, but this requires a comprehensive revision.

Thank you for noting our commitment to transparency regarding uncertainties in our study. We take utmost seriousness in openly reporting uncertainties and informing readers about any issues they should be aware of. Therefore, we believe that the length and comprehensiveness of the article are justified. Acknowledging the uncertainties, our conclusions are appropriately cautious. We do not assert that the soil carbon stock increased with high confidence; rather, we state that the estimated soil carbon influx and efflux did not definitively indicate a reduction in soil carbon stock within the study sites (L622-623). Utilizing literature values would compromise the study's objective of assessing site-specific carbon balance representing Baltic states. Additionally, employing heterotrophic respiration measurements or literature values would introduce further uncertainty, as discussed both presently and in the article. Therefore, we believe that a full re-analysis is not necessary; however, we will do our best to improve the clarity of the text, including motivation for the choices that were criticized, while shortening and streamlining the actual manuscript by placing some sections in supplementary materials. For instance, the selection of boreal or temperate literature values for comparison is based on the IPCC classification, which places the Baltic states in the temperate zone. Therefore, for GHG inventories in these countries, temperate default emission factors should be applied until higher tier factors are available. However, Kottek et.all. (2006) indicated that most of the area of the Baltic states would rather be in the boreal zone, we will add this reasoning

We appreciate your valuable feedback and understand the concerns raised. We believe that providing clarifications based on the information already included in the article may address most of these concerns without requiring extensive revisions of the paper. Most of the mentioned aspects can be clarified with rather minor technical edits. Our goal is to ensure that the article provides convenient reading while still providing the necessary details for a comprehensive understanding of our study and the results obtained.

35: Other studies exist that present the C balance between heterotrophic C loss from peat and inputs from litter in forested peatlands (Hermans et al 2022).

We agree. The statement is widely accepted knowledge, so we considered that excessive referencing is unnecessary. However, we can include this reference.

54-79: This paragraph gives a lengthy account of technical considerations for C accounting under the IPCC. It does not focus sufficiently on the scientific background and goals f the research. Whilst I appreciate the consideration for harmonised protocols and potential of bias from using contrasting schemes, this should be referenced or presented much more concisely to maintain a focus on the advancement of peatland drainage understanding of C balances per se, not technicalities in its reporting.

We believe that it is crucial to include this information to engage a broader readership, including greenhouse gas inventory specialists, and provide a comprehensive context for them.

While the primary purpose of the article undeniably aims to enhance the scientific understanding of carbon balance, we equally value dedicating attention to the practical application of the findings. This is especially pertinent due to growing concerns among various stakeholders, including policymakers and greenhouse gas inventory compilers, regarding organic soils. We also wish to point out that "ordinary" readers often look for some specific details in papers that they are reviewing, and we would like our paper to be as useful as possible in this respect. We agree that information of scientific relevance of the study is beneficial to the section and we will be happy to expand it. However, we can also move some issues to Supplementary materials, if deemed useful.

84-86: The hypothesis is not statistically testable. Of course, consistent emission factors can be used (tailored or not), but there is no statistical method to accept or refute such a statement. Please present an actual hypothesis 9that can be phrased as a null hypothesis, i.e. is statistically testable). As it is presented, the objective seems to be to collect and p[resent emissions data that can be used in future analyses – why is it not being analysed or synthesised here?

The hypothesis is statistically tested by comparing emissions stratified by dominant tree species or countries using pairwise Wilcoxon rank sum tests (L267). Additionally, we assess the significance of the country impact on emissions through mixed-effects models (L260). Yet, we will happily consider sharpening the aims statement and hypotheses following these comments, when revising the manuscript.

The data is analysed here, but with the additional wish that the article provides an extensive amount of data suitable for separate use in future studies or for combining with additional results to synthesize new findings. We advocate for open science principles and, accordingly, we also provide the full data used in carbon balance estimations to enhance its future utilization. We are personally rather frustrated by papers that seem to aim to telling a well-flowing story instead of providing the necessary details to be wider applicable by the scientific community.

104 (Table 1; small detail): Where n = 2, please just state the respective values separated by comma, rather than "…", which implies a wider range of values.

Thank you for noting this, we will correct this issue.

120-121: The second sentence of the paragraph seems to repeat the exact information given in the first sentence.

Indeed, thank you for noticing this, the issue will be corrected.

122; Hutchinson and Livingston (1993) describe opaque chamber methods, but you should provide specific detail of your chamber dimensions.

The chamber dimensions are specified in L122-123, with a volume of 0.0655 m³ and an area of 0.196 m². Consequently, the chambers were relatively large, which likely explains why we did not observe the impact of pressure increase on gas concentration, as discussed earlier.

126: Heterotrophic decomposition of soil organic matter (peat) is surely also included!

That is correct, here we simply emphasise that also autotrophic respiration and litter layer decomposition is included. For clarity we will add "…not only soil heterotrophic respiration but also…".

132: 30 or 60-minute sampling intervals will lead to significant build-up of CO2 in chambers with likely non-linear diffusion flux. Using a simple linear regression is likely to under-estimate flux values. See e.g. Kutzbach et al. (2007; Biogeosciences, 4(6), pp.1005-1025). The degree of under-estimation is likely to be dependent on the degree of concentration build-up (i.e. flux magnitude).

We believe the concern is resolved by answers provided above.

135: This is not right. An ECD can not detect CO2.

ECD can detect CO2 (e.g. Ferraz-Almeida et.al. (2020), Loftfield (1997, Maier et.al. (2022))

148: Please clarify if the heterotrophic respiration measurements were taken over the same periods and on the same days as the main flux measurements. Why is a different system used for these measurements?

Heterotrophic respiration and total respiration measurements were done during the same study site survey – same day and same time periods. The approach was chosen during the initial harmonisation of methodology between study teams of Estonia, Latvia, and Lithuania lead by colleagues from Finland (LUKE). Final decision was made to use larger chambers (area 0.196 m², volume 0.0655 m³) for measuring total respiration with gas chromatograph (measurement time 30 or 60 minutes) and smaller chambers (area 0.07 m², volume 0.017 m³) to measure heterotrophic respiration using NDIR (measurement time 3 minutes) to harmonize approaches also with previous studies for comparability reasons. Larger chamber for total respiration is needed to enable entrapment of ground vegetation in the chamber headspace.

212: Not clear: Rhet measurements were made on litter-free soil with no root influence, so you should use different C input values compared to Rtot calculations.

We would have used different C input values if directly measured Rhet results would have been usable. However, because the used Rhet derived from Rtot (L224-228) for the reasons discussed earlier, a unified approach for C input values was deemed appropriate to employ.

219-223: Predicting soil C output using abiotic drivers is not trivial. And you have to provide significantly more information regarding the underlying regression used. From Table S6, I gather that you used a linear temperature response, which is unusual, as there is abundant literature to show that respiration follows an exponential temperature response. Soil moisture is an additional factor, and its influence should be investigated in combination with temperature. Potential underestimation bias from the concentration build-up in chambers is likely to affect higher fluxes during warmer conditions more strongly than colder/low flux conditions, resulting in a more liner response.

We fully agree. Initially, three separate approaches were employed to derive regressions: nonlinear, linear after log-transformation, and linear after Box-Cox data transformation (L258-265). The Box-Cox data transformation provided the best prediction (L343-345), hence linear models were deemed appropriate and are presented in Table S6. Previous studies have

indicated that the Box-Cox approach helps mitigate biases introduced by nonlinearity (references at L262, L527). While we believe the derivation of models is transparent, we can clarify in the "Statistical analysis" section that three approaches were initially considered, and the linear fit after Box-Cox transformation was found to be the most suitable. These models were subsequently used for flux interpolation. We acknowledge the importance of soil moisture as discussed in L466-469 and L477-480 and reported in articles referred in L527. However, our study did not include continuous soil moisture measurements to address this issue.

319: You appear to treat each chamber measurement as an independent observation, but as the locations were identical for each plot, you should account for this temporal pseudo-replication by applying repeated-measures statistics.

The locations were not identical but reflect local diversity in vegetation and potentially groundwater levels along the transect of sampling sites (L114). What has traditionally been called pseudo-replication (and sometimes indeed incorrectly utilized) provides in fact critical information of measurement site heterogeneity, and can be utilized.

340-344: Did you attempt to use an exponential response, rather than linear regression?

Linear regressions were applied to transformed data as discussed above. We did try also exponential response, but linear fit after data transformation provided better performance of the models (L343-344).

370-377: It is unclear why there should be any difference in the correlation between Rtot or Rhet and different soil parameters, as one is derived directly from the other, so correlations should be equally as strong – or in any case, they are not independent from one another. Figure S6 shows correlation results for sil depths of "0 – 30 cm", but the text references soil depth of 20-30 cm only.

The keyword here is "measured Rhet" not derived Rhet, for this reason the correlations differ. Thank you for noticing this, we will correct the reference by moving it to appropriate location of the paragraph. We will also introduce separate terms for measured and calculated Rhet for clarity.

544-546: This is also very unclear. What are "removals" by NEE observed in hemiboreal zones? Provide a reference and make it clear if this refers to a higher rate of NPP. Higher NPP relates to pretty much all ecosystem components, not just litter. And finally: are "Rhet C loss" and "Rhet rates" not the same flux?

According to Krasnova et al. (2019), NEE tends to be more negative in the hemiboreal zone compared to the boreal zone, indicating net carbon sequestration by the ecosystem. This is likely facilitated by higher net primary productivity (NPP), leading to increased biomass litterfall and consequently higher carbon input into the soil, which counteracts the relatively higher heterotrophic respiration rates observed.

We acknowledge the need for clarity in the statement regarding 'Rhet C loss' and 'Rhet rates'. The intention was to highlight that higher Rhet rates in the hemiboreal zone may also be associated with higher NPP. To prevent misunderstandings, we will revise the sentence to: 'One reason for this is the larger removals by net ecosystem exchange observed in the hemiboreal

zone compared to northern forests, ensured by higher gross primary productivity (GPP) (Krasnova et al., 2019), which creates a greater potential for carbon influx through litter to offset soil carbon loss from Rhet.'

597-601: Rather than speculating about whether the public finds results controversial (where is the evidence or motivation for this statement?), you should present robust interpretation of what can be concluded. The figures you cite show fluxes not significantly different from zero, so whilst they don't support findings of soils being a c source, you can also not present them as a C sink.

Thank you, we will check the phrasing (at least) one more time for this. The uncertainties are reported and allow for the interpretation that while mean carbon balance indicates removals those removals are highly uncertain; therefore we do not conclude that the soil is a sink, we conclude that we did not find evidence for soil C loss, please refer to the section Conclusions. For clarity we can improve the sentence as follows: "The acquired empirical data segregated by drainage status indicated that neither drained nor undrained nutrient rich organic forest soils in Baltic states showed a C source, but rather minor C removals of $0.9\pm1.51$ t C ha−1 year−1 and $1.19\pm1.48$ t C ha−1 year−1, respectively. Judged further by the uncertainty around the mean values, it can be concluded that the soils appear to be close to C neutral".

608-610: This is unclear. What is the assumed C stock 100 years ago for this assertion? And why do you apply the temperate emissions factor when otherwise comparing to boreal or hemiboreal conditions in the manuscript

We can consider to either remove the assertion or clarify that for this assertion we assume C stock before drainage was similar to one measured in the undrained sites currently and add reference to studies assessing the C stocks in similar drained and undrained sites in the region. Temperate emission factors are applied due to IPCC guidelines mandating their use in the absence of country-specific factors for the Baltic states, despite evidence indicating that the conditions align more closely with the boreal zone. According to updated KÖPPEN-GEIGER climate classification Baltic States align with boreal zone. Also, previous studies show that soil GHG emissions align more with the ones elaborated in boreal zone rather than in Temperate zone. Consequently, while scientifically it would be more valid to compare Baltic states with boreal zone rather than a temperate zone, in the scope of GHG inventories methodology we are mandated to be compared with temperate zone. For this reason, in specific context, we compare the results also with the temperate emission factors. Such approach underscores the added value of the article, as it promotes a shift from broad regional emission factors to those more appropriate for local conditions, enhancing the accuracy of soil carbon balance predictions.

833 (Figure S4): The units on they-axis seem wrong as values can not represent fluxes in mg C m-2 h-1.

Unit is in ln transformed mg C m-2 h-1, we will improve clarity by adding this information also in the caption.

References used:

Alisa Krasnova, Mai Kukumägi, Ülo Mander, Raili Torga, Dmitrii Krasnov, Steffen M. Noe, Ivika Ostonen, Ülle Püttsepp, Helen Killian, Veiko Uri, Krista Lõhmus, Jaak Sõber, Kaido Soosaar. (2019) Carbon exchange in a hemiboreal mixed forest in relation to tree species composition. Agricultural and Forest Meteorology, Volume 275, 2019, Pages 11-23, ISSN 0168-1923, https://doi.org/10.1016/j.agrformet.2019.05.007

Ferraz-Almeida, R., Spokas, K. A., & De Oliveira, R. C. (2020). Columns and Detectors Recommended in Gas Chromatography to Measure Greenhouse Emission and O2 Uptake in Soil: A Review. Communications in Soil Science and Plant Analysis, 51(5), 582–594. https://doi.org/10.1080/00103624.2020.1729370

Jauhiainen, J., Alm, J., Bjarnadottir, B., Callesen, I., Christiansen, J., Clarke, N., Dalsgaard, L., He, H., Jordan, S., Kazanavičiūtė, V., Klemedtsson, L., Lauren, A., Lazdins, A., Lehtonen, A., Lohila, A., Lupikis, A., Mander, U., Minkkinen, K., Kasimir, ., Olsson, M., Ojanen, P., Oskarsson, H., Sigurdsson, B., Søgaard, G., Soosaar, K., Vesterdal, L., & Laiho, R. (2019). Reviews and syntheses: Greenhouse gas exchange data from drained organic forest soils – a review of current approaches and recommendations forfuture research. Biogeosciences, 16(23), 4687–4703.

Jauhiainen, J., Heikkinen, J., Clarke, N., He, H., Dalsgaard, L., Minkkinen, K., Ojanen, P., Vesterdal, L., Alm, J., Butlers, A., Callesen, I., Jordan, S., Lohila, A., Mander, U., Oskarsson, H., Sigurdsson, B., Søgaard, G., Soosaar, K., Kasimir, ., Bjarnadottir, B., Lazdins, A., & Laiho, R. (2023). Reviews and syntheses: Greenhouse gas emissions from drained organic forestsoils – synthesizing data for site-specific emission factors for boreal andcool temperate regions. Biogeosciences, 20(23), 4819–4839.

Kottek, M., J. Grieser, C. Beck, B. Rudolf, and F. Rubel, 2006: World Map of the Köppen-Geiger climate classification updated. Meteorol. Z., 15, 259-263. DOI: 10.1127/0941-2948/2006/0130.

Loftfield, N., Flessa, H., Beese, F., & Augustin, J. (1997). Automated Gas Chromatographic System for Rapid Analysis of the Atmospheric Trace Gases Methane, Carbon Dioxide, and Nitrous Oxide. Journal of Environmental Quality - J ENVIRON QUAL, 26, 560-564.

Maier M. *et al.* Introduction of a guideline for measurements of greenhouse gas fluxes from soils using non-steady-state chambers, J. Plant Nutr. Soil Sci., vol. 185, no. 4, 2022, pp. 447–461, DOI: 10.1002/jpln.202200199

---

## Author Comment (AC2)

The paper is targeting the important question of CO2-emissions from drained forested peatlands and their potentially higher emissions than undrained forested peatlands. This is important in the context of GHG reporting (UNFCCC). They establish extensive systems for measurements during a 2-year period in a total of 26 sites (n=19 are drained) distributed to all three Baltic countries. They underline the uncertainty of the current default IPCC emission factor for the region and that the transition between temperate and boreal zone (hemiboreal) may be poorly represented by the current IPCC default. Thus, I very much welcome and acknowledge the effort and recognize its importance. Particularly in light of future policy demands on the LULUCF sector emissions/sequestrations and the need to increase Tier levels and enhance documentation.

It is my opinion that the paper still needs some work to be ready for publication. This relates to the clarity in methods, thorough discussion of uncertainties including reference to magnitudes and drivers found in other studies (Rtot, Rhet), the use of extensive soil chemical data and on the text and priorities of both introduction and discussion. See specific comments below.

**Thank you for the positive view, the very constructive comments, and the in-depth review of our article! Your comments are well-received and will greatly help us improve the article.**

Specific comments:

The introduction largely refers to IPCC guidelines and national scale emission estimates based on UNFCCC (national submissions). I miss an effort to link the study to existing research literature on what the important drivers have been found to be, if emissions are mainly climate-driven or if they have also been found to be driven by vegetation, history, geology/landscape etc. It is stated that earlier results are "inconclusive" but you should at least mention what other studies have expected to find and what was concluded. A paragraph on how forested peatlands in the Baltic region may differ from those used in the IPCC default EF as well as the potential methodologies would be interesting in this context. The study focuses on nutrient rich sites. While the study has an apparent aim to contribute to higher certainty for the UNFCCC reporting for the Baltic countries, it is not shown to what extent the selected sites are area-representative for the drained peatlands for which these countries need to report.

**We agree that more strongly addressing the scientific relevance of the study, including the current scientific understanding and knowledge gaps, can be beneficial. We will consider how to incorporate all the listed aspects in the Introduction.The availability of previous research results on the key drivers of organic soil $CO_2$ emissions allows for their association with climate, land use history, and soil nutrient status, provided that the amount of studies in the respective climate zone permits it (Hiraishi et al. 2013). While the IPCC (2014) evaluated soil nutrient in the boreal zone only, the ammount of studies conducted in the last decade allows for further EF stratification by categories of afforested sites and forestry-drained sites, as well as more specific site nutrient status and productivity categories based on timber production potential also in the temperate zone (Jauhiainen et al. 2023).The primitive approach clearly indicates necessity to seek for broader understandement. Boreal vegetation zone has the most extensive and consistent data; however, in the temperate climate zone (including hemiboreal vegetation zone), the amount of research results remains significantly lower, resulting in a greater uncertainty (Jauhiainen et al. 2019, 2023). The contradictions in research results are likely due to this uncertainty, which has prevented a consensus on whether organic soil is a**

source or sink. It has been observed that forest soils in moist and cool climates do not necessarily lose carbon stock. Studies show that drained organic forest soils C stock is stable or even increasing (Butlers et al. 2022; Minkkinen et al. 2018), particularly in nutrient-poor or moderate nutrient-rich soil (Lazdiņš et al. 2024). Nonetheless, there are observations that a higher risk of soil C loss can be observed in mostly nutrient-rich soils, often showing a C loss (Lazdiņš et al. 2024). However, observations indicate that C stock of such soils can also be preserved (Butlers et al. 2022). The available data is still insufficient to confidently identify the key influencing factors contributing to the inconclusiveness of these results. This is largely due to the specific local conditions of the study sites, such as meteorological conditions, water table level, vegetation, and soil properties. Quantifying the impact of the variability of these conditions on soil carbon balance remains complex.

An emission increase gradient from Nordic countries to Central Europe can be observed based on earlier studies. This climatic gradient also explains the differences in peatlands in the Baltic states. Vegetation, soil properties, temperature, and groundwater level dynamics are all influenced by climate, which is why the peatlands in the Baltic states differ from those characterized by the IPCC default emission factor. The IPCC factor represents geographic regions with warmer and drier climates.

We will also deal with the site representativeness issue in the Material and Methods section, and briefly return to it in Discussion for improved transparency. We focused on nutrient-rich soils because they are associated with higher emissions; it seems rational to begin improving the accuracy of such soil emissions assessments. By applying the obtained knowledge to all areas, we can avoid underestimating emissions while later working on understanding nutrient-poor soils.

Methods should include the history of the sites studied (I don't find this), particularly when they were drained (and perhaps drainage channels were maintained over time as a typical management activity through time), their LU/area characteristics before planting (if they were planted). What does one know in terms of expecting that these forested peatlands were similar to the undrained forested peatland that are included in the study? The undrained sites in general show a higher tree basal area than the drained sites – do the undrained sites represent sites that would have been selected to be drained historically? In the paper you mention several places the "effect" of drainage. I claim that you are not measuring an effect of drainage, but you are comparing (contrast) two types of forest with apparent different management over time – and most likely the drainage happened long ago.

Thank you for your detailed observations. We agree that information about historical land use would be beneficial for transparent scientific reporting and facilitating interpretation. Unfortunately, historical land use is often poorly documented or not documented at all in the Baltic states. While we intended to include such information, it is unfortunately unavailable. We can, however, provide a general description of typical land use history in this region, and include the best possible evaluation of what has been done previously at our study sites. Water table level can be elevated in drained areas and lowered in undrained areas. To characterise the functioning of drainage system and soil moisture conditions we aimed to describe our observations regarding water table levels in detail. This will help indicate whether existing drainage systems are functioning and what the water table characteristics are. We observed that in undrained areas, the average measured water table level was above -30 cm. However, the drained study sites showed a wide gradient of average water table levels, ranging from

approximately -30 cm to more than -100 cm. Extensive drainage was conducted around the mid-20th century. The drainage was typically not performed for afforestation but to improve tree growth conditions in already forested areas. This means that forest development in all areas (both undrained and drained) occurred naturally as a natural succession initially. Due to the aforementioned reasons: these undrained sites included in the study can be expected to be similar to histroical forested peatlands; it is unlikely that the currently undrained areas were historically drained; and undrained sites represent sites that would have been selected to be drained as the water level was consistently elevated.

Indeed, the lower end of the basal area range is higher in undrained sites, which coincides with similar observations in forest stand ages. Such observation arises due to sites selected and does not imply that basal areas of undrained sites tend to be higher in undrained sites. We will consider how to concisely include this descriptive information in the article.

We agree that the term "effect" can be reconsidered, and comparisons between undrained and drained sites should be interpreted carefully here, given that the drainage occurred long ago. We can revise the phrasing to clearly indicate that we are comparing the two groups of sites, and that at best, we may be evaluating the long-term effect of drainage based on assumed similarity before drainage.

Also, sites likely do vary a little in ground vegetation composition – it would be timely to have a clear description of ground vegetation as you use only some of the vegetation components in the balance calculations.

That is correct, ground vegetation is typically different in drained and undrained areas, and there is, naturally, at least some variation also within the site groups. We have conducted vegetation surveys at sites in Latvia, so we can provide at least descriptive information to illustrate the vegetation present in these areas.

Method description of respiration (total, heterotrophic), litter input fluxes and C balances (forest floor, soil) need to be supplemented by a figure with the fluxes that are measured and estimated and how they are combined to calculate the soil and forest floor balances. I believe this will make the methods much more clear as well as shorten the text.

There is such a figure already elaborated, we will consider if it helps the readers, and check the potential to shorten the text while improving the clarity of method description using the figure.

You observe that measured Rhet results seem unreliable, very high compared to Rtot and with a poor correlation to temperature r2 < 0.3) relative to Rtot (r2 ca. 0.7-0.8). In some context (fx. line 462) you mention they are found to be in error. In other context you claim that they are unlikely to be subject to measurement errors but mention their likely influence from decomposing roots (line 499), or the lack of temperature and moisture measurements that reflect the actual measurement position (poor correlation to temperature, unknown potential effects of moisture fluctuations). As far as I can see you i) describe the field measurement methods for Rhet in detail, ii) discard the results (line 214) for use in the C balances, iii) do not present them in the Results but iv) refer to them with correlations with Corg%, C:N, porosity

or BD (line 436 and onward…however, it is unclear to me if your reference to Rhet here is to the measured Rhet or the Rhet used in the C balance calculations (eq 2)). I realize you wish to present openly to the reader what you have done and which problems you encountered. I feel the balance in the paper of this challenge is wrong. I think what I would do would be: include the field method description of Rhet in the main text if your results from these measurements are still helping you in your research aims. If not – I would move most of it to the supplement.

I would use more effort when selecting the empirical relationship by selecting (reviewing) more than one. The chosen one is from boreal forest (you claim in the intro that hemiboreal EF is likely not represented by temperate EF, back up your choice of a Rhet estimation regression from a boreal study). If I understand correctly that you refer to measured Rhet in the discussion (correlations with Corg%, C:N, porosity or BD) I would like to know why you believe this is relevant given the likely effects of decomposing roots on measured Rhet. Given the clear effect that your decision has on not using the measured Rhet in C-balances I would like to see in the main text a figure with magnitudes and correlations for Rhet and Rtot (clarifying to the reader) and an opportunity to clearly state what you use the measured Rhet for and what not. What are the correlations to temperature and the magnitudes one can expect? And please use different abbreviations for the measured Rhet and the estimated Rhet from eq. 2. An uncertainty discussion should include the uncertainties inherent in the choice of using the eq2 (alternative Rhet).

Thank you for this observation, and the very helpful comments on how to improve the clarity and motivation of our choices. We now clearly see that there remained some confusion in the text relative to the challenges that we faced with measured Rhet, and the need to improve the clarity of presenting these findings. We will gladly follow the suggestions provided. You are correct in understanding that the measured Rhet results were not used in the C balance calculation, but they were used to describe specific relationships between emissions and soil quality indicators. We have outlined this explanation in L437-438, but it is evident that we should also include it in the sections where we describe Rhet (measurements rather than estimations) and their influencing factors.

We can improve the clarity in the article that these influencing factors were more closely correlated with the measured Rhet than with Rtot. Although the measured Rhet results were not otherwise utilized, we aimed to highlight potential uses of soil properties for characterizing soil emissions.

Since the direct Rhet measurements were not used in quantifying the C balance, we agree that some of this information can be moved to the supplementary material. We believe it is essential to inform the reader about this critical Rhet measurement aspect of our study by providing a brief outline in the main text to indicate that these observations were made, but the results were not used in the C balance calculation due to significant issues that justified this decision. A detailed explanation of how the measurements were taken and the associated problems can be moved to the supplementary material. This approach will help avoid overloading the main article while still informing the reader about these important observations.

Thank you for the suggestion to use a different term for the calculated Rhet. We will implement this change. We can indeed also add a figure with the magnitudes and correlations for Rhet and Rtot, and dedicate a subsection in the results for a concise Rhet description, including the current quantitative information from the discussion, while moving the rest to the

supplementary materials. The Rhet overestimation compared to the Rtot data can be presented, e.g., in a boxplot format. We will also include a note about the quality issues of the Rhet measurements. We struggled with how to present the measured Rhet data in a meaningful way that would both add critical information and transparency in the paper and support future studies on these issues. We believe that the revision will lead to fulfilling both these aims.

Both results and discussion use considerable space on describing observed effects/relationships between fluxes and soil nutrient characteristics. I miss a much more clear direction on these tests and on the discussion of their results and this direction should be set in the introduction, preferably as specific research questions and/or hypotheses.

We will consider how to improve the presentation of these results. It is unlikely that we will formulate a specific hypothesis regarding soil quality, as we performed these analyses to characterize the study site conditions, while evaluating soil quality as such was not our aim. The opportunity to compare these results with emissions adds value as an attempt to find factors correlating with the fluxes and explaining variation among sites. We could describe this as our intention, to collect data to evaluate the potential use of such data as predictors for soil emissions in the future. Specifically, in this case, to assess the relationships between soil quality and emissions.

It is clear that studies of this scale will not yield specific results applicable in GHG reporting on their own. Even if practical applications were identified, they would not be feasible when detailed soil quality maps do not exist. However, they might exist in the future. Therefore, this contribution is an ongoing effort to gather data from study to study, describing what is observed in smaller studies, and later working towards more concrete solutions by combining the results of this and other studies. We could mention this in the introduction to explain our interest presenting these results, e.g., in the new section suggested for evaluating what is known about the drivers of the fluxes.

Application of statistical tests (methods) are not clearly described in terms of testing expected biological relationships; every test should be used for a clear purpose. Given the few sites some of the statistical methods rely on very few observations per strata. I think it would be better to limit analyses to the mixed model analyses. I find that the PCA analyses are not utilized to their potential – fx. one could use the PC vectors as explanatory variables and – if they express environmental variability that is possible to interpret in a meaningful way – they may help to find a pattern in how the many measured variables influence/drive emissions (example: Callesen et al. 2006. Growth of Beech, Oak, and Four Conifer Species Along a Soil Fertility Gradient. Baltic Forestry Vol. 12, No. 1 (22)).

We will revise the description of the statistical analyses, adding the specific purpose of each test. We agree that the observation count is indeed small in some cases; however, we used different approaches that provided confluences towards building observations and conclusions. Therefore, we considered it valuable to show that different methods provide the same indications of relationships. However, we will reevaluate this carefully, and may move some confirmative tests and their results to supplementary material.

We used PCA in primarily to  visualize the covariation in the data, and to evaluate whether there were any clear patterns of country or dominant tree species affecting soil emissions or other quantitative characteristics of the study sites. One of our aims was to evaluate if dominant species or country should be used to stratify emission factors. Ultimately, the confluences of results from various statistical methods led to the conclusion that it would not be scientifically justified, and a single emission factor should be used for all drained or undrained soils. We agree that PCA could be used further for evaluating the relationships between the variables. We will carefully consider the suggestion in the context of this article. Thank you for providing the literature.

The discussion starts by targeting the errors observed in one of the flux-methods (unclear if Rhet is the measured one or the one from eq. 2). Rather, I believe the discussion should start by referring to the results actually obtained in the study on the outcomes you have targeted in your study aim (probably the balance rather than any specific flux?). And target uncertainties on specific fluxes (parts of the balance) in later sections. As an example, in the very last sentence of the section on soil heterotrophic respiration interpretation you state that roots cut in the process of installation was most likely the major reason for your errors…if this is the most important contribution then you should start the section on soil heterotrophic respiration error-discussions with this. Also, I would expect that you would find several studies who have found a similar challenge and it would be timely to refer to such.

We agree that the logical flow can be improved here. Indeed, there are previous studies discussing trenching issues, and we will seek to include some references to show that similar discussions have been raised before. As previously agreed, based on your valuable suggestions, discussions related to directly measured Rhet will likely be moved to the supplementary materials. We agree that the article should focus on data directly used in C balance estimation. However, since the Rhet-related findings are likely valuable for some of the readers, we will include in the article the reasoning for data exclusions, with a note indicating that a more in-depth discussion on this issue can be found in the supplementary materials.

Thank you for providing many suggestions for technical corrections. We genuinely appreciate the time you have dedicated to helping make our article as readable and accessible as possible for the reader. We will address all of these suggestions.